# Exploiting Higher Order Smoothness in Derivative-free Optimization and Continuous Bandits

**Arya Akhavan**
Istituto Italiano di Tecnologia
and
ENSAE Paris Tech
aria.akhavanfoomani@iit.it

**Massimiliano Pontil**
Istituto Italiano di Tecnologia
and
University College London
massimiliano.pontil@iit.it

**Alexandre B. Tsybakov**
ENSAE Paris Tech
alexandre.tsybakov@ensae.fr

## Abstract

We study the problem of zero-order optimization of a strongly convex function. The goal is to find the minimizer of the function by a sequential exploration of its values, under measurement noise. We study the impact of higher order smoothness properties of the function on the optimization error and on the cumulative regret. To solve this problem we consider a randomized approximation of the projected gradient descent algorithm. The gradient is estimated by a randomized procedure involving two function evaluations and a smoothing kernel. We derive upper bounds for this algorithm both in the constrained and unconstrained settings and prove minimax lower bounds for any sequential search method. Our results imply that the zero-order algorithm is nearly optimal in terms of sample complexity and the problem parameters. Based on this algorithm, we also propose an estimator of the minimum value of the function achieving almost sharp oracle behavior. We compare our results with the state-of-the-art, highlighting a number of key improvements.

## 1   Introduction

We study the problem of zero-order stochastic optimization, in which we aim to minimize an unknown strongly convex function via a sequential exploration of its function values, under measurement error, and a closely related problem of continuous (or continuum-armed) stochastic bandits. These problems have received significant attention in the literature, see [1, 2, 3, 4, 7, 9, 10, 14, 17, 18, 34, 16, 20, 21, 30, 25, 31, 13, 27, 28, 19, 29], and are fundamental for many applications in which the derivatives of the function are either too expensive or impossible to compute. A principal goal of this paper is to exploit higher order smoothness properties of the underlying function in order to improve the performance of search algorithms. We derive upper bounds on the estimation error for a class of projected gradient-like algorithms, as well as close matching lower bounds, that characterize the role played by the number of iterations, the strong convexity parameter, the smoothness parameter, the number of variables, and the noise level.

Let $f : \mathbb{R}^d \to \mathbb{R}$ be the function that we wish to minimize over a closed convex subset $\Theta$ of $\mathbb{R}^d$. Our approach, outlined in Algorithm 1, builds upon previous work in which a sequential algorithm queries at each iteration a pair of function values, under a general noise model. Specifically, at iteration $t$ the current guess $x_t$ for the minimizer of $f$ is used to build two perturbations $x_t + \delta_t$ and $x_t - \delta_t$, where the function values are queried subject to additive measurement errors $\xi_t$ and $\xi'_t$, respectively. The

---

**Algorithm 1** Zero-Order Stochastic Projected Gradient

---

**Requires** Kernel $K : [-1, 1] \to \mathbb{R}$, step size $\eta_t > 0$ and parameter $h_t$, for $t = 1, \dots, T$

**Initialization** Generate scalars $r_1, \dots, r_T$ uniformly on the interval $[-1, 1]$, vectors $\zeta_1, \dots, \zeta_T$ uniformly distributed on the unit sphere $S_d = \{\zeta \in \mathbb{R}^d : \|\zeta\| = 1\}$, and choose $x_1 \in \Theta$

**For** $t = 1, \dots, T$

    1. Let $y_t = f(x_t + h_t r_t \zeta_t) + \xi_t$ and $y'_t = f(x_t - h_t r_t \zeta_t) + \xi'_t$,

    2. Define $\hat{g}_t = \frac{d}{2h_t}(y_t - y'_t)\zeta_t K(r_t)$

    3. Update $x_{t+1} = \mathrm{Proj}_\Theta(x_t - \eta_t \hat{g}_t)$

**Return** $(x_t)_{t=1}^T$

---

values $\delta_t$ can be chosen in different ways. In this paper, we set $\delta_t = h_t r_r \zeta_t$ (Line 1), where $h_t > 0$ is a suitably chosen small parameter, $r_t$ is random and uniformly distributed on $[-1, 1]$, and $\zeta_t$ is uniformly distributed on the unit sphere. The estimate for the gradient is then computed at Line 2 and used inside a projected gradient method scheme to compute the next exploration point. We introduce a suitably chosen kernel $K$ that allows us to take advantage of higher order smoothness of $f$.

The idea of using randomized procedures for derivative-free stochastic optimization can be traced back to Nemirovski and Yudin [23, Sec. 9.3] who suggested an algorithm with one query per step at point $x_t + h_t \zeta_t$, with $\zeta_t$ uniform on the unit sphere. Its versions with one, two or more queries were studied in several papers including [1, 3, 16, 31]. Using two queries per step leads to better performance bounds as emphasized in [26, 1, 3, 16, 31, 13]. Randomizing sequences other than uniform on the sphere were also explored: $\zeta_t$ uniformly distributed on a cube [26], Gaussian $\zeta_t$ [24, 25], $\zeta_t$ uniformly distributed on the vertices of a cube [30] or satisfying some general assumptions [12, 13]. Except for [26, 12, 3], these works study settings with low smoothness of $f$ (2-smooth or less) and do not invoke kernels $K$ (i.e. $K(\cdot) \equiv 1$ and $r_t \equiv 1$ in Algorithm 1). The use of randomization with smoothing kernels was proposed by Polyak and Tsybakov [26] and further developed by Dippon [12], and Bach and Perchet [3] to whom the current form of Algorithm 1 is due.

In this paper we consider higher order smooth functions $f$ satisfying the generalized Hölder condition with parameter $\beta \geq 2$, cf. inequality (1) below. For integer $\beta$, this parameter can be roughly interpreted as the number of bounded derivatives. Furthermore, we assume that $f$ is $\alpha$-strongly convex. For such functions, we address the following two main questions:

(a) What is the performance of Algorithm 1 in terms of the cumulative regret and optimization error, namely what is the explicit dependency of the rate on the main parameters $d, T, \alpha, \beta$?

(b) What are the fundamental limits of any sequential search procedure expressed in terms of minimax optimization error?

To handle task (a), we prove upper bounds for Algorithm 1, and to handle (b), we prove minimax lower bounds for any sequential search method.

**Contributions.** Our main contributions can be summarized as follows: **i)** Under an adversarial noise assumption (cf. Assumption 2.1 below), we establish for all $\beta \geq 2$ upper bounds of the order $\frac{d^2}{\alpha} T^{-\frac{\beta-1}{\beta}}$ for the optimization risk and $\frac{d^2}{\alpha} T^{\frac{1}{\beta}}$ for the cumulative regret of Algorithm 1, both for its constrained and unconstrained versions; **ii)** In the case of independent noise satisfying some natural assumptions (including the Gaussian noise), we prove a minimax lower bound of the order $\frac{d}{\alpha} T^{-\frac{\beta-1}{\beta}}$ for the optimization risk when $\alpha$ is not very small. This shows that to within the factor of $d$ the bound for Algorithm 1 cannot be improved for all $\beta \geq 2$; **iii)** We show that, when $\alpha$ is too small, below some specified threshold, higher order smoothness does not help to improve the convergence rate. We prove that in this regime the rate cannot be faster than $d/\sqrt{T}$, which is not better (to within the dependency on $d$) than for derivative-free minimization of simply convex functions [2, 18]; **iv)** For $\beta = 2$, we obtain a bracketing of the optimal rate between $O(d/\sqrt{\alpha T})$ and $\Omega(d/(\max(1, \alpha)\sqrt{T}))$. In a special case when $\alpha$ is a fixed numerical constant, this validates a conjecture in [30] (claimed there as proved fact) that the optimal rate for $\beta = 2$ scales as $d/\sqrt{T}$; **v)** We propose a simple algorithm of estimation of the value $\min_x f(x)$ requiring three queries per step and attaining the optimal rate $1/\sqrt{T}$ for all

$\beta \geq 2$. The best previous work on this problem [6] suggested a method with exponential complexity and proved a bound of the order $c(d, \alpha)/\sqrt{T}$ for $\beta > 2$ where $c(d, \alpha)$ is an unspecified constant.

**Notation.** Throughout the paper we use the following notation. We let $\langle \cdot, \cdot \rangle$ and $\| \cdot \|$ be the standard inner product and Euclidean norm on $\mathbb{R}^d$, respectively. For every close convex set $\Theta \subset \mathbb{R}^d$ and $x \in \mathbb{R}^d$ we denote by $\text{Proj}_{\Theta}(x) = \operatorname{argmin}\{\|z - x\| : z \in \Theta\}$ the Euclidean projection of $x$ to $\Theta$. We assume everywhere that $T \geq 2$. We denote by $\mathcal{F}_{\beta}(L)$ the class of functions with Hölder smoothness $\beta$ (inequality (1) below). Recall that $f$ is $\alpha$-strongly convex for some $\alpha > 0$ if, for any $x, y \in \mathbb{R}^d$ it holds that $f(y) \geq f(x) + \langle \nabla f(x), y - x \rangle + \frac{\alpha}{2}\|x - y\|^2$. We further denote by $\mathcal{F}_{\alpha,\beta}(L)$ the class of all $\alpha$-strongly convex functions belonging to $\mathcal{F}_{\beta}(L)$.

**Organization.** We start in Section 2 with some preliminary results on the gradient estimator. Section 3 presents our upper bounds for Algorithm 1, both in the constrained and unconstrained case. In Section 4 we observe that a slight modification of Algorithm 1 can be used to estimated the minimum value (rather than the minimizer) of $f$. Section 4 presents improved upper bounds in the case $\beta = 2$. In Section 6 we establish minimax lower bounds. Finally, Section 7 contrasts our results with previous work in the literature and discusses future directions of research.

## 2 Preliminaries

In this section, we give the definitions, assumptions and basic facts that will be used throughout the paper. For $\beta > 0$, let $\ell$ be the greatest integer strictly less than $\beta$. We denote by $\mathcal{F}_{\beta}(L)$ the set of all functions $f : \mathbb{R}^d \to \mathbb{R}$ that are $\ell$ times differentiable and satisfy, for all $x, z \in \Theta$ the Hölder-type condition

$$\left| f(z) - \sum_{0 \leq |m| \leq \ell} \frac{1}{m!} D^m f(x)(z - x)^m \right| \leq L\|z - x\|^{\beta}, \tag{1}$$

where $L > 0$, the sum is over the multi-index $m = (m_1, ..., m_d) \in \mathbb{N}^d$, we used the notation $m! = m_1! \cdots m_d!$, $|m| = m_1 + \cdots + m_d$, and we defined

$$D^m f(x)\nu^m = \frac{\partial^{|m|} f(x)}{\partial^{m_1} x_1 \cdots \partial^{m_d} x_d} \nu_1^{m_1} \cdots \nu_d^{m_d}, \quad \forall \nu = (\nu_1, \ldots, \nu_d) \in \mathbb{R}^d.$$

In this paper, we assume that the gradient estimator defined by Algorithm 1 uses a kernel function $K : [-1, 1] \to \mathbb{R}$ satisfying

$$\int K(u)du = 0, \int uK(u)du = 1, \int u^j K(u)du = 0, \ j = 2, \ldots, \ell, \ \int |u|^{\beta}|K(u)|du < \infty. \tag{2}$$

Examples of such kernels obtained as weighted sums of Legendre polynomials are given in [26] and further discussed in [3].

**Assumption 2.1.** *It holds, for all $t \in \{1, \ldots, T\}$, that: (i) the random variables $\xi_t$ and $\xi_t'$ are independent from $\zeta_t$ and from $r_t$, and the random variables $\zeta_t$ and $r_t$ are independent; (ii) $\mathbb{E}[\xi_t^2] \leq \sigma^2$, and $\mathbb{E}[(\xi_t')^2] \leq \sigma^2$, where $\sigma \geq 0$.*

Note that we do not assume $\xi_t$ and $\xi_t'$ to have zero mean. Moreover, they can be non-random and no independence between noises on different steps is required, so that the setting can be considered as adversarial. Having such a relaxed set of assumptions is possible because of randomization that, for example, allows the proofs go through without assuming the zero mean noise.

We will also use the following assumption.

**Assumption 2.2.** *Function $f : \mathbb{R}^d \to \mathbb{R}$ is 2-smooth, that is, differentiable on $\mathbb{R}^d$ and such that $\|\nabla f(x) - \nabla f(x')\| \leq \bar{L}\|x - x'\|$ for all $x, x' \in \mathbb{R}^d$, where $\bar{L} > 0$.*

It is easy to see that this assumption implies that $f \in \mathcal{F}_2(\bar{L}/2)$. The following lemma gives a bound on the bias of the gradient estimator.

**Lemma 2.3.** *Let $f \in \mathcal{F}_{\beta}(L)$, with $\beta \geq 1$ and let Assumption 2.1 (i) hold. Let $\hat{g}_t$ and $x_t$ be defined by Algorithm 1 and let $\kappa_{\beta} = \int |u|^{\beta}|K(u)|du$. Then*

$$\|\mathbb{E}[\hat{g}_t \,|\, x_t] - \nabla f(x_t)\| \leq \kappa_{\beta}Ldh_t^{\beta-1}. \tag{3}$$

If $K$ be a weighted sum of Legendre polynomials, $\kappa_\beta \leq 2\sqrt{2}\beta$, with $\beta \geq 1$ (see e.g., [3, Appendix A.3]).

The next lemma provides a bound on the stochastic variability of the estimated gradient by controlling its second moment.

**Lemma 2.4.** *Let Assumption 2.1(i) hold, let $\hat{g}_t$ and $x_t$ be defined by Algorithm 1 and set $\kappa = \int K^2(u)du$. Then*

*(i) If $\Theta \subseteq \mathbb{R}^d$, $\nabla f(x^*) = 0$ and Assumption 2.2 holds,*

$$\mathbb{E}[\|\hat{g}_t\|^2 \,|\, x_t] \leq 9\kappa \bar{L}^2 \left( d\|x_t - x^*\|^2 + \frac{d^2 h_t^2}{8} \right) + \frac{3\kappa d^2 \sigma^2}{2h_t^2},$$

*(ii) If $f \in \mathcal{F}_2(L)$ and $\Theta$ is a closed convex subset of $\mathbb{R}^d$ such that $\max_{x \in \Theta}\|\nabla f(x)\| \leq G$, then*

$$\mathbb{E}[\|\hat{g}_t\|^2 \,|\, x_t] \leq 9\kappa \left( G^2 d + \frac{L^2 d^2 h_t^2}{2} \right) + \frac{3\kappa d^2 \sigma^2}{2h_t^2}.$$

## 3  Upper bounds

In this section, we provide upper bounds on the cumulative regret and on the optimization error of Algorithm 1, which are defined as

$$\sum_{t=1}^{T} \mathbb{E}[f(x_t) - f(x)],$$

and

$$\mathbb{E}[f(\hat{x}_T) - f(x^*)],$$

respectively, where $x \in \Theta$ and $\hat{x}_T$ is an estimator after $T$ queries. Note that the provided upper bound for cumulative regret is valid for any $x \in \Theta$.

First we consider Algorithm 1 when the convex set $\Theta$ is bounded (constrained case).

**Theorem 3.1.** (Upper Bound, Constrained Case.) *Let $f \in \mathcal{F}_{\alpha,\beta}(L)$ with $\alpha, L > 0$ and $\beta \geq 2$. Let Assumptions 2.1 and 2.2 hold and let $\Theta$ be a convex compact subset of $\mathbb{R}^d$. Assume that $\max_{x \in \Theta}\|\nabla f(x)\| \leq G$. If $\sigma > 0$ then the cumulative regret of Algorithm 1 with*

$$h_t = \left( \frac{3\kappa \sigma^2}{2(\beta - 1)(\kappa_\beta L)^2} \right)^{\frac{1}{2\beta}} t^{-\frac{1}{2\beta}}, \quad \eta_t = \frac{2}{\alpha t}, \quad t = 1, \ldots, T$$

*satisfies*

$$\forall x \in \Theta : \sum_{t=1}^{T} \mathbb{E}[f(x_t) - f(x)] \leq \frac{1}{\alpha} \left( d^2 \left( A_1 T^{1/\beta} + A_2 \right) + A_3 d \log T \right), \quad (4)$$

*where $A_1 = 3\beta(\kappa\sigma^2)^{\frac{\beta-1}{\beta}}(\kappa_\beta L)^{\frac{2}{\beta}}$, $A_2 = \bar{c}\bar{L}^2(\sigma/L)^{\frac{2}{\beta}} + 9\kappa G^2/d$ with constant $\bar{c} > 0$ depending only on $\beta$, and $A_3 = 9\kappa G^2$. The optimization error of averaged estimator $\bar{x}_T = \frac{1}{T}\sum_{t=1}^{T} x_t$ satisfies*

$$\mathbb{E}[f(\bar{x}_T) - f(x^*)] \leq \frac{1}{\alpha} \left( d^2 \left( \frac{A_1}{T^{\frac{\beta-1}{\beta}}} + \frac{A_2}{T} \right) + A_3 \frac{d \log T}{T} \right), \quad (5)$$

*where $x^* = \arg\min_{x \in \Theta} f(x)$. If $\sigma = 0$, then the cumulative regret and the optimization error of Algorithm 1 with any $h_t$ chosen small enough and $\eta_t = \frac{2}{\alpha t}$ satisfy the bounds (4) and (5), respectively, with $A_1 = 0$, $A_2 = 9\kappa G^2/d$ and $A_3 = 10\kappa G^2$.*

**Proof sketch.** We use the definition of Algorithm 1 and strong convexity of $f$ to obtain an upper bound for $\sum_{t=1}^{T} \mathbb{E}[f(x_t) - f(x)|x_t]$, which depends on the bias term $\sum_{t=1}^{T} \|\mathbb{E}[\hat{g}_t \,|\, x_t] - \nabla f(x_t)\|$ and on the stochastic error term $\sum_{t=1}^{T} \mathbb{E}[\|\hat{g}_t\|^2]$. By substituting $h_t$ (that is derived from balancing the two terms) and $\eta_t$ in Lemmas 2.3 and 2.4 we obtain upper bounds for $\sum_{t=1}^{T} \|\mathbb{E}[\hat{g}_t \,|\, x_t] - \nabla f(x_t)\|$ and $\sum_{t=1}^{T} \mathbb{E}[\|\hat{g}_t\|^2]$ that imply the desired upper bound for $\sum_{t=1}^{T} \mathbb{E}[f(x_t) - f(x)|x_t]$ due to a recursive argument in the spirit of [5]. ∎

In the non-noisy case ($\sigma = 0$) we get the rate $\frac{d}{\alpha} \log T$ for the cumulative regret, and $\frac{d}{\alpha} \frac{\log T}{T}$ for the optimization error. In what concerns the optimization error, this rate is not optimal since one can achieve much faster rate under strong convexity [25]. However, for the cumulative regret in our derivative-free setting it remains an open question whether the result of Theorem 3.1 can be improved. Previous papers on derivative-free online methods with no noise [1, 13, 16] provide slower rates than $(d/\alpha) \log T$. The best known so far is $(d^2/\alpha) \log T$, cf. [1, Corollary 5]. We may also notice that the cumulative regret bounds of Theorem 3.1 trivially extend to the case when we query functions $f_t$ depending on $t$ rather than a single $f$. Another immediate fact is that on the r.h.s. of inequalities (4) and (5) we can take the minimum with $GBT$ and $GB$, respectively, where $B$ is the Euclidean diameter of $\Theta$. Finally, the factor $\log T$ in the bounds for the optimization error can be eliminated by considering averaging from $T/2$ to $T$ rather than from $1$ to $T$, in the spirit of [27]. We refer to Appendix D for the details and proofs of these facts.

We now study the performance of Algorithm 1 when $\Theta = \mathbb{R}^d$. In this case we make the following choice for the parameters $h_t$ and $\eta_t$ in Algorithm 1:

$$
\begin{aligned}
h_t &= T^{-\frac{1}{2\beta}}, \quad \eta_t = \frac{1}{\alpha T}, \quad t = 1, \ldots, T_0, \\
h_t &= t^{-\frac{1}{2\beta}}, \quad \eta_t = \frac{2}{\alpha t}, \quad t = T_0 + 1, \ldots, T,
\end{aligned}
\tag{6}
$$

where $T_0 = \max\left\{k \geq 0 : C_1 \bar{L}^2 d > \alpha^2 k / 2\right\}$ and $C_1$ is a positive constant[1] depending only on the kernel $K(\cdot)$ (this is defined in the proof of Theorem 3.2 in Appendix B) and recall $\bar{L}$ is the Lipschitz constant on the gradient $\nabla f$. Finally, define the estimator

$$
\bar{x}_{T_0,T} = \frac{1}{T - T_0} \sum_{t=T_0+1}^{T} x_t.
\tag{7}
$$

**Theorem 3.2.** (Upper Bounds, Unconstrained Case.) *Let $f \in \mathcal{F}_{\alpha,\beta}(L)$ with $\alpha, L > 0$ and $\beta \geq 2$. Let Assumptions 2.1 and 2.2 hold. Assume also that $\alpha > \sqrt{C_* d/T}$, where $C_* > 72\kappa \bar{L}^2$. Let $x_t$'s be the updates of Algorithm 1 with $\Theta = \mathbb{R}^d$, $h_t$ and $\eta_t$ as in (6) and a non-random $x_1 \in \mathbb{R}^d$. Then the estimator defined by (7) satisfies*

$$
\mathbb{E}[f(\bar{x}_{T_0,T}) - f(x^*)] \leq C\kappa\bar{L}^2 \frac{d}{\alpha T} \|x_1 - x^*\|^2 + C\frac{d^2}{\alpha}\left((\kappa_\beta L)^2 + \kappa(\bar{L}^2 + \sigma^2)\right)T^{-\frac{\beta-1}{\beta}}
\tag{8}
$$

*where $C > 0$ is a constant depending only on $\beta$ and $x^* = \arg\min_{x \in \mathbb{R}^d} f(x)$.*

**Proof sketch.** As in the proof of Theorem 3.1, we apply Lemmas 2.3 and 2.4. But we can only use Lemma 2.4(i) and not Lemma 2.4(ii) and thus the bound on the stochastic error now involves $\|x_t - x^*\|^2$. So, after taking expectations, we need to control an additional term containing $r_t = \mathbb{E}[\|x_t - x^*\|^2]$. However, the issue concerns only small $t$ ($t \leq T_0 \sim d^2/\alpha$) since for bigger $t$ this term is compensated due to the strong convexity with parameter $\alpha > \sqrt{C_* d/T}$. This motivates the method where we use the first $T_0$ iterations to get a suitably good (but not rate optimal) bound on $r_{T_0+1}$ and then proceed analogously to Theorem 3.1 for iterations $t \geq T_0 + 1$. ∎

## 4 Estimation of $f(x^*)$

In this section, we apply the above results to estimation of the minimum value $f(x^*) = \min_{x \in \Theta} f(x)$ for functions $f$ in the class $\mathcal{F}_{\alpha,\beta}(L)$. The literature related to this problem assumes that $x_t$'s are either i.i.d. with density bounded away from zero on its support [32] or $x_t$'s are chosen sequentially [22, 6]. In the fist case, from the results in [32] one can deduce that $f(x^*)$ cannot be estimated better than at the slow rate $T^{-\beta/(2\beta+d)}$. For the second case, which is our setting, the best result so far is obtained in [6]. The estimator of $f(x^*)$ in [6] is defined via a multi-stage procedure whose complexity increases exponentially with the dimension $d$ and it is shown to achieve (asymptotically,

for $T$ greater than an exponent of $d$) the $c(d, \alpha)/\sqrt{T}$ rate for functions in $\mathcal{F}_{\alpha,\beta}(L)$ with $\beta > 2$. Here, $c(d, \alpha)$ is some constant depending on $d$ and $\alpha$ in an unspecified way.

Observe that $f(\bar{x}_T)$ is not an estimator since it depends on the unknown $f$, so Theorem 3.1 does not provide a result about estimation of $f(x^*)$. In this section, we show that using the computationally simple Algorithm 1 and making one more query per step (that is, having three queries per step in total) allows us to achieve the $1/\sqrt{T}$ rate for all $\beta \geq 2$ with no dependency on the dimension in the main term. Note that the $1/\sqrt{T}$ rate cannot be improved. Indeed, one cannot estimate $f(x^*)$ with a better rate even using the ideal but non-realizable oracle that makes all queries at point $x^*$. That is, even if $x^*$ is known and we sample $T$ times $f(x^*) + \xi_t$ with independent centered variables $\xi_t$, the error is still of the order $1/\sqrt{T}$.

In order to construct our estimator, at any step $t$ of Algorithm 1 we make along with $y_t$ and $y_t'$ the third query $y_t'' = f(x_t) + \xi_t''$, where $\xi_t''$ is some noise and $x_t$ are the updates of Algorithm 1. We estimate $f(x^*)$ by $\hat{M} = \frac{1}{T}\sum_{t=1}^{T} y_t''$. The properties of estimator $\hat{M}$ are summarized in the next theorem, which is an immediate corollary of Theorem 3.1.

**Theorem 4.1.** *Let the assumptions of Theorem 3.1 be satisfied. Let $\sigma > 0$ and assume that $(\xi_t'')_{t=1}^{T}$ are independent random variables with $\mathbb{E}[\xi_t''] = 0$ and $\mathbb{E}[(\xi_t'')^2] \leq \sigma^2$ for $t = 1, \ldots, T$. If $f$ attains its minimum at point $x^* \in \Theta$, then*

$$\mathbb{E}|\hat{M} - f(x^*)| \leq \frac{\sigma}{T^{\frac{1}{2}}} + \frac{1}{\alpha}\left(d^2\left(\frac{A_1}{T^{\frac{\beta-1}{\beta}}} + \frac{A_2}{T}\right) + A_3\frac{d\log T}{T}\right). \tag{9}$$

**Remark 4.2.** *With three queries per step, the risk (error) of the oracle that makes all queries at point $x^*$ does not exceed $\sigma/\sqrt{3T}$. Thus, for $\beta > 2$ the estimator $\hat{M}$ achieves asymptotically as $T \to \infty$ the oracle risk up to a numerical constant factor. We do not obtain such a sharp property for $\beta = 2$, in which case the remainder term in Theorem 4.1 accounting for the accuracy of Algorithm 1 is of the same order as the main term $\sigma/\sqrt{T}$.*

Note that in Theorem 4.1 the noises $(\xi_t'')_{t=1}^{T}$ are assumed to be independent and zero mean random variables, which is essential to obtain the $1/\sqrt{T}$ rate. Nevertheless, we do not require independence between the noises $(\xi_t'')_{t=1}^{T}$ and the noises in the other two queries $(\xi_t)_{t=1}^{T}$ and $(\xi_t')_{t=1}^{T}$. Another interesting point is that for $\beta = 2$ the third query is not needed and $f(x^*)$ is estimated with the $1/\sqrt{T}$ rate either by $\hat{M} = \frac{1}{T}\sum_{t=1}^{T} y_t$ or by $\hat{M} = \frac{1}{T}\sum_{t=1}^{T} y_t'$. This is an easy consequence of the above argument, the property (19) – see Lemma A.3 in the appendix – which is specific for the case $\beta = 2$, and the fact that the optimal choice of $h_t$ is of order $t^{-1/4}$ for $\beta = 2$.

## 5 Improved bounds for $\beta = 2$

In this section, we consider the case $\beta = 2$ and obtain improved bounds that scale as $d$ rather than $d^2$ with the dimension in the constrained optimization setting analogous to Theorem 3.1. First note that for $\beta = 2$ we can simplify the algorithm. The use of kernel $K$ is redundant when $\beta = 2$, and therefore in this section we define the approximate gradient as

$$\hat{g}_t = \frac{d}{2h_t}(y_t - y_t')\zeta_t, \tag{10}$$

where $y_t = f(x + h_t\tilde{\zeta})$ and $y_t' = f(x - h_t\tilde{\zeta})$. A well-known observation that goes back to [23] consists in the fact that $\hat{g}_t$ defined in (10) is an unbiased estimator of the gradient of the surrogate function $\hat{f}_t$ defined by

$$\hat{f}_t(x) = \mathbb{E}f(x + h_t\tilde{\zeta}), \quad \forall x \in \mathbb{R}^d,$$

where the expectation $\mathbb{E}$ is taken with respect to the random vector $\tilde{\zeta}$ uniformly distributed on the unit ball $B_d = \{u \in \mathbb{R}^d : \|u\| \leq 1\}$. The properties of the surrogate $\hat{f}_t$ are described in Lemmas A.2 and A.3 presented in the appendix.

The improvement in the rate that we get for $\beta = 2$ is due to the fact that we can consider Algorithm 1 with $\hat{g}_t$ defined in (10) as the SGD for the surrogate function. Then the bias of approximating $f$ by $\hat{f}_t$ scales as $h_t^2$, which is smaller than the squared bias of approximating the gradient arising in the proof

of Theorem 3.1 that scales as $d^2 h_t^{2(\beta-1)} = d^2 h_t^2$ when $\beta = 2$. On the other hand, the stochastic variability terms are the same for both methods of proof. This explains the gain in dependency on $d$. However, this technique does not work for $\beta > 2$ since then the error of approximating $f$ by $\hat{f}_t$, which is of the order $h_t^\beta$ (with $h_t$ small), becomes too large compared to the bias $d^2 h_t^{2(\beta-1)}$ of Theorem 3.1.

**Theorem 5.1.** *Let $f \in \mathcal{F}_{\alpha,2}(L)$ with $\alpha, L > 0$. Let Assumption 2.1 hold and let $\Theta$ be a convex compact subset of $\mathbb{R}^d$. Assume that $\max_{x \in \Theta} \|\nabla f(x)\| \leq G$. If $\sigma > 0$ then for Algorithm 1 with $\hat{g}_t$ defined in (10) and parameters $h_t = \left( \frac{3d^2 \sigma^2}{4L\alpha t + 9L^2 d^2} \right)^{1/4}$ and $\eta_t = \frac{1}{\alpha t}$ we have*

$$\forall x \in \Theta: \quad \mathbb{E} \sum_{t=1}^{T} \left( f(x_t) - f(x) \right) \leq \min \left( GBT, 2\sqrt{3L}\sigma \frac{d}{\sqrt{\alpha}} \sqrt{T} + A_4 \frac{d^2}{\alpha} \log T \right), \quad (11)$$

*where $B$ is the Euclidean diameter of $\Theta$ and $A_4 = 6.5L\sigma + 22G^2/d$. Moreover, if $x^* = \arg\min_{x \in \Theta} f(x)$ the optimization error of averaged estimator $\bar{x}_T = \frac{1}{T} \sum_{t=1}^{T} x_t$ is bounded as*

$$\mathbb{E}[f(\bar{x}_T) - f(x^*)] \leq \min \left( GB, 2\sqrt{3L}\sigma \frac{d}{\sqrt{\alpha T}} + A_4 \frac{d^2}{\alpha} \frac{\log T}{T} \right). \quad (12)$$

*Finally, if $\sigma = 0$, then the cumulative regret of Algorithm 1 with any $h_t$ chosen small enough and $\eta_t = \frac{1}{\alpha t}$ and the optimization error of its averaged version are of the order $\frac{d^2}{\alpha} \log T$ and $\frac{d^2}{\alpha} \frac{\log T}{T}$, respectively.*

Note that the terms $\frac{d^2}{\alpha} \log T$ and $\frac{d^2}{\alpha} \frac{\log T}{T}$ appearing in these bounds can be improved to $\frac{d}{\alpha} \log T$ and $\frac{d}{\alpha} \frac{\log T}{T}$ at the expense of assuming that the norm $\|\nabla f\|$ is uniformly bounded by $G$ not only on $\Theta$ but also on a large enough Euclidean neighborhood of $\Theta$. Moreover, the $\log T$ factor in the bounds for the optimization error can be eliminated by considering averaging from $T/2$ to $T$ rather than from 1 to $T$ in the spirit of [27]. We refer to Appendix D for the details and proofs of these facts. A major conclusion is that, when $\sigma > 0$ and we consider the optimization error, those terms are negligible with respect to $d/\sqrt{\alpha T}$ and thus an attainable rate is $\min(1, d/\sqrt{\alpha T})$.

We close this section by noting, in connection with the bandit setting, that the bound (11) extends straightforwardly (up to a change in numerical constants) to the cumulative regret of the form $\mathbb{E} \sum_{t=1}^{T} \left( f_t(x_t \pm h_t \zeta_t) - f_t(x) \right)$, where the losses are measured at the query points and $f$ depends on $t$. This fact follows immediately from the proof of Theorem 5.1 presented in the appendix and the property (19), see Lemma A.3 in the appendix.

# 6 Lower bound

In this section we prove a minimax lower bound on the optimization error over all sequential strategies that allow the query points depend on the past. For $t = 1, \ldots, T$, we assume that $y_t = f(z_t) + \xi_t$ and we consider strategies of choosing the query points as $z_t = \Phi_t(z_1^{t-1}, y_1^{t-1})$ where $\Phi_t$ are Borel functions and $z_1 \in \mathbb{R}^d$ is any random variable. We denote by $\Pi_T$ the set of all such strategies. The noises $\xi_1, \ldots, \xi_T$ are assumed in this section to be independent with cumulative distribution function $F$ satisfying the condition

$$\int \log \left( dF(u)/dF(u+v) \right) dF(u) \leq I_0 v^2, \quad |v| < v_0 \quad (13)$$

for some $0 < I_0 < \infty, 0 < v_0 \leq \infty$. Using the second order expansion of the logarithm w.r.t. $v$, one can verify that this assumption is satisfied when $F$ has a smooth enough density with finite Fisher information. For example, for Gaussian distribution $F$ this condition holds with $v_0 = \infty$. Note that the class $\Pi_T$ includes the sequential strategy of Algorithm 1 that corresponds to taking $T$ as an even number, and choosing $z_t = x_t + \zeta_t r_t$ and $z_t = x_t - \zeta_t r_t$ for even $t$ and odd $t$, respectively. The presence of the randomizing sequences $\zeta_t, r_t$ is not crucial for the lower bound. Indeed, Theorem 6.1 below is valid conditionally on any randomization, and thus the lower bound remains valid when taking expectation over the randomizing distribution.

**Theorem 6.1.** *Let* $\Theta = \{x \in \mathbb{R}^d : \|x\| \leq 1\}$. *For* $\alpha, L > 0, \beta \geq 2$, *let* $\mathcal{F}'_{\alpha,\beta}$ *denote the set of functions* $f$ *that attain their minimum over* $\mathbb{R}^d$ *in* $\Theta$ *and belong to* $\mathcal{F}_{\alpha,\beta}(L) \cap \{f : \max_{x \in \Theta} \|\nabla f(x)\| \leq G\}$, *where* $G > 2\alpha$. *Then for any strategy in the class* $\Pi_T$ *we have*

$$\sup_{f \in \mathcal{F}'_{\alpha,\beta}} \mathbb{E}\big[f(z_T) - \min_x f(x)\big] \geq C \min\Big(\max(\alpha, T^{-1/2+1/\beta}), \frac{d}{\sqrt{T}}, \frac{d}{\alpha}T^{-\frac{\beta-1}{\beta}}\Big), \qquad (14)$$

*and*

$$\sup_{f \in \mathcal{F}'_{\alpha,\beta}} \mathbb{E}\big[\|z_T - x^*(f)\|^2\big] \geq C \min\Big(1, \frac{d}{T^{\frac{1}{\beta}}}, \frac{d}{\alpha^2}T^{-\frac{\beta-1}{\beta}}\Big), \qquad (15)$$

*where* $C > 0$ *is a constant that does not depend of* $T$, $d$, *and* $\alpha$, *and* $x^*(f)$ *is the minimizer of* $f$ *on* $\Theta$.

The proof is given in Appendix B. It extends the proof technique of Polyak and Tsybakov [28], by applying it to more than two probe functions. The proof takes into account dependency on the dimension $d$, and on $\alpha$. The final result is obtained by applying Assouad's Lemma, see e.g. [33].

We stress that the condition $G > 2\alpha$ in this theorem is necessary. It should always hold if the intersection $\mathcal{F}_{\alpha,\beta}(L) \cap \{f : \max_{x \in \Theta} \|\nabla f(x)\| \leq G\}$ is not empty. Notice also that the threshold $T^{-1/2+1/\beta}$ on the strong convexity parameter $\alpha$ plays an important role in bounds (14) and (15). Indeed, for $\alpha$ below this threshold, the bounds start to be independent of $\alpha$. Moreover, in this regime, the rate of (14) becomes $\min(T^{1/\beta}, d)/\sqrt{T}$, which is asymptotically $d/\sqrt{T}$ and thus not better as function of $T$ than the rate attained for zero-order minimization of simply convex functions [2, 7]. Intuitively, it seems reasonable that $\alpha$-strong convexity should be of no added value for very small $\alpha$. Theorem 6.1 allows us to quantify exactly how small such $\alpha$ should be. Also, quite naturally, the threshold becomes smaller when the smoothness $\beta$ increases.

Finally note that for $\beta = 2$ the lower bounds (14) and (15) are, in the interesting regime of large enough $T$, of order $d/(\max(\alpha, 1)\sqrt{T})$ and $d/(\max(\alpha^2, 1)\sqrt{T})$, respectively. This highlights the near minimax optimal properties of Algorithm 1 in the setting of Theorem 5.1.

## 7  Discussion and related work

There is a great deal of attention to zero-order feedback stochastic optimization and convex bandits problems in the recent literature. Several settings are studied: (i) deterministic in the sense that the queries contain no random noise and we query functions $f_t$ depending on $t$ rather than $f$ where $f_t$ are Lipschitz or 2-smooth [16, 1, 24, 25, 28, 31]; (ii) stochastic with two-point feedback where the two noisy evaluations are obtained with the same noise and the noisy functions are Lipschitz or 2-smooth [24, 25, 13] (this setting does not differ much from (i) in terms of the analysis and the results); (iii) stochastic, where the noises $\xi_i$ are independent zero-mean random variables [15, 26, 12, 2, 30, 3, 19, 4, 20]. In this paper, we considered a setting, which is more general than (iii) by allowing for adversarial noise (no independence or zero-mean assumption in contrast to (iii), no Lipschitz assumption in contrast to settings (i) and (ii)), which are both covered by our results when the noise is set to zero.

One part of our results are bounds on the cumulative regret, cf. (4) and (11). We emphasize that they remain trivially valid if the queries are from $f_t$ depending on $t$ instead of $f$, and thus cover the setting (i). To the best of our knowledge, there were no such results in this setting previously, except for [3] that gives bounds with suboptimal dependency on $T$ in the case of classical (non-adversarial) noise. In the non-noisy case, we get bounds on the cumulative regret with faster rates than previously known for the setting (i). It remains an open question whether these bounds can be improved.

The second part of our results dealing with the optimization error $\mathbb{E}[f(\bar{x}_T) - f(x^*)]$ is closely related to the work on derivative-free stochastic optimization under strong convexity and smoothness assumptions initiated in [15, 26] and more recently developed in [12, 19, 30, 3]. It was shown in [26] that the minimax optimal rate for $f \in \mathcal{F}_{\alpha,\beta}(L)$ scales as $c(\alpha, d)T^{-(\beta-1)/\beta}$, where $c(\alpha, d)$ is an unspecified function of $\alpha$ and $d$ (for $d = 1$ an upper bound of the same order was earlier established in [15]). The issue of establishing non-asymptotic fundamental limits as function of the main parameters of the problem ($\alpha$, $d$ and $T$) was first addressed in [19] giving a lower bound $\Omega(\sqrt{d/T})$ for $\beta = 2$. This was improved to $\Omega(d/\sqrt{T})$ when $\alpha \asymp 1$ by Shamir [30] who conjectured that the rate $d/\sqrt{T}$ is optimal for $\beta = 2$, which indeed follows from our Theorem 5.1 (although [30] claims the optimality

as proved fact by referring to results in [1], such results cannot be applied in setting (iii) because the noise cannot be considered as Lipschitz). A result similar to Theorem 5.1 is stated without proof in Bach and Perchet [3, Proposition 7] but not for the cumilative regret and with a suboptimal rate in the non-noisy case. For integer $\beta \geq 3$, Bach and Perchet [3] present explicit upper bounds as functions of $\alpha$, $d$ and $T$ with, however, suboptimal dependency on $T$ except for their Proposition 8 that is problematic (see Appendix C for the details). Finally, by slightly modifying the proof of Theorem 3.1 we get that the estimation risk $\mathbb{E}\big[\left\|\bar{x}_T - x^*\right\|^2\big]$ is $O((d^2/\alpha^2)T^{-(\beta-1)/\beta})$, which is to within factor $d$ of the main term in the lower bound (15) (see Appendix D for details).

The lower bound in Theorem 6.1 is, to the best of our knowledge, the first result providing non-asymptotic fundamental limits under general configuration of $\alpha$, $d$ and $T$. The known lower bounds [26, 19, 30] either give no explicit dependency on $\alpha$ and $d$, or treat the special case $\beta = 2$ and $\alpha \asymp 1$. Moreover, as an interesting consequence of our lower bound we find that, for small strong convexity parameter $\alpha$ (namely, below the $T^{-1/2+1/\beta}$ threshold), the best achievable rate cannot be substantially faster than for simply convex functions, at least for moderate dimensions. Indeed, for such small $\alpha$, our lower bound is asymptotically $\Omega(d/\sqrt{T})$ independently of the smoothness index $\beta$ and on $\alpha$, while the achievable rate for convex functions is shown to be $d^{16}/\sqrt{T}$ in [2] and improved to $d^{3.75}/\sqrt{T}$ in [7] (both up to log-factors). The gap here is only in the dependency on the dimension. Our results imply that for $\alpha$ above the $T^{-1/2+1/\beta}$ threshold, the gap between upper and lower bounds is much smaller. Thus, our upper bounds in this regime scale as $(d^2/\alpha)T^{-(\beta-1)/\beta}$ while the lower bound of Theorem 6.1 is of the order $\Omega\big((d/\alpha)T^{-(\beta-1)/\beta}\big)$; moreover for $\beta = 2$, upper and lower bounds match in the dependency on $d$.

We hope that our work will stimulate further study at the intersection of zero-order optimization and convex bandits in machine learning. An important open problem is to study novel algorithms which match our lower bound simultaneously in all main parameters. For example a class of algorithms worth exploring are those using memory of the gradient in the spirit of Nesterov accelerated method. Yet another important open problem is to study lower bounds for the regret in our setting. Finally, it would be valuable to study extensions of our work to locally strongly convex functions.

## Broader impact

The present work improves our understanding of zero-order optimization methods in specific scenarios in which the underlying function we wish to optimize has certain regularity properties. We believe that a solid theoretical foundation is beneficial to the development of practical machine learning and statistical methods. We expect no direct or indirect ethical risks from our research.

## Acknowledgments and Disclosure of Funding

We would like to thank Francis Bach, Vianney Perchet, Saverio Salzo, and Ohad Shamir for helpful discussions. The first and second authors were partially supported by SAP SE. The research of A.B. Tsybakov is supported by a grant of the French National Research Agency (ANR), "Investissements d'Avenir" (LabEx Ecodec/ANR-11-LABX-0047).

## Footnotes

[1]If $T_0 = 0$ the algorithm does not use (6). Assumptions of Theorem 3.2 are such that condition $T > T_0$ holds.

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
