[Supplementary Material]

# Supplementary material

The supplementary material is organized as follows. In Appendix A we provide some auxiliary results, including those stated in Section 2 above. In Appendix B we give proofs of the results which were only stated or whose proof was only sketched in the paper. For reader's convenience all such results are restated below. Appendix C contains some comments on previous results in [3]. Finally, in Appendix D we present refined versions of Theorems 3.1 and 5.1.

## A   Auxiliary results

**Lemma 2.3.** *Let $f \in \mathcal{F}_\beta(L)$, with $\beta \geq 1$ and let Assumption 2.1 (i) hold. Let $\hat{g}_t$ and $x_t$ be defined by Algorithm 1 and let $\kappa_\beta = \int |u|^\beta |K(u)| du$. Then*

$$\|\mathbb{E}[\hat{g}_t \,|\, x_t] - \nabla f(x_t)\| \leq \kappa_\beta L d h_t^{\beta-1}. \tag{3}$$

**Proof.** To lighten the presentation and without loss of generality we drop the lower script "$t$" in all quantities. Using the Taylor expansion we have

$$f(x + hr\zeta) = f(x) + \langle \nabla f(x), hr\zeta \rangle + \sum_{2 \leq |m| \leq \ell} \frac{(rh)^{|m|}}{m!} D^{(m)} f(x) \zeta^m + R(hr\zeta),$$

where by assumption $|R(hr\zeta)| \leq L\|hr\zeta\|^\beta = L|r|^\beta h^\beta$. Thus,

$$\mathbb{E}[\hat{g}|x] = \frac{d}{h} \mathbb{E}\Big[\Big(\langle \nabla f(x), hr\zeta \rangle + \sum_{2 \leq |m| \leq \ell, |m| \text{ odd}} \frac{(rh)^{|m|}}{m!} D^{(m)} f(x) \zeta^m + \frac{R(hr\zeta) - R(-hr\zeta)}{2}\Big) \zeta K(r)\Big].$$

Since $\zeta$ is uniformly distributed on the unit sphere we have $\mathbb{E}[\zeta\zeta^\top] = (1/d)I_{d \times d}$, where $I_{d \times d}$ is the identity matrix. Therefore,

$$\mathbb{E}\Big[\frac{d}{h} \langle \nabla f(x), h\zeta \rangle \zeta\Big] = \nabla f(x).$$

As $\int r^{|m|} K(r) dr = 0$ for $2 \leq |m| \leq \ell$ and $\int r K(r) dr = 1$ we conclude that

$$
\begin{aligned}
\|\mathbb{E}[\hat{g}\,|\,x] - \nabla f(x)\| &= \frac{d}{2h} \|\mathbb{E}\big[\big(R(hr\zeta) - R(-hr\zeta)\big) \zeta K(r)\big]\| \\
&\leq \frac{d}{2h} \mathbb{E}\big[|R(hr\zeta) - R(-hr\zeta)|\,|K(r)|\big] \leq \kappa_\beta L d h^{\beta-1}.
\end{aligned}
$$

∎

**Lemma 2.4.** *Let Assumption 2.1(i) hold, let $\hat{g}_t$ and $x_t$ be defined by Algorithm 1 and set $\kappa = \int K^2(u) du$. Then*

*(i) If $\Theta \subseteq \mathbb{R}^d$, $\nabla f(x^*) = 0$ and Assumption 2.2 holds,*

$$\mathbb{E}[\|\hat{g}_t\|^2 \,|\, x_t] \leq 9\kappa \bar{L}^2 \Big(d\|x_t - x^*\|^2 + \frac{d^2 h_t^2}{8}\Big) + \frac{3\kappa d^2 \sigma^2}{2h_t^2},$$

*(ii) If $f \in \mathcal{F}_2(L)$ and $\Theta$ is a closed convex subset of $\mathbb{R}^d$ such that $\max_{x \in \Theta} \|\nabla f(x)\| \leq G$, then*

$$\mathbb{E}[\|\hat{g}_t\|^2 \,|\, x_t] \leq 9\kappa \Big(G^2 d + \frac{L^2 d^2 h_t^2}{2}\Big) + \frac{3\kappa d^2 \sigma^2}{2h_t^2}.$$

**Proof.** We have

$$
\begin{aligned}
\|\hat{g}\|^2 &= \frac{d^2}{4h^2} \big\|\big(f(x + hr\zeta) - f(x - hr\zeta) + \xi - \xi'\big)\zeta K(r)\big\|^2 \\
&= \frac{d^2}{4h^2}\big(f(x + hr\zeta) - f(x - hr\zeta) + \xi - \xi'\big)^2 K^2(r).
\end{aligned}
$$

Using the inequality $(a + b + c)^2 \leq 3(a^2 + b^2 + c^2)$ we get

$$\mathbb{E}[\|\hat{g}\|^2 \,|\, x] \leq \frac{3d^2}{4h^2} \left( \mathbb{E}\left[ (f(x + hr\zeta) - f(x - hr\zeta))^2 K^2(r) \right] + 2\kappa\sigma^2 \right). \tag{16}$$

Here,

$$
\begin{aligned}
\big(f(x + hr\zeta) - f(x - hr\zeta)\big)^2 &= \big(f(x + hr\zeta) - f(x - hr\zeta) \pm f(x) \pm 2\langle \nabla f(x), hr\zeta \rangle\big)^2 \\
&\leq 3\bigg\{ \Big( f(x + hr\zeta) - f(x) - \langle \nabla f(x), hr\zeta \rangle \Big)^2 \\
&\quad + \Big( f(x - hr\zeta) - f(x) - \langle \nabla f(x), -hr\zeta \rangle \Big)^2 + 4\langle \nabla f(x), hr\zeta \rangle^2 \bigg\} \\
&\leq 3\left( \frac{\bar{L}^2}{2} \|hr\zeta\|^4 + 4\langle \nabla f(x), hr\zeta \rangle^2 \right),
\end{aligned}
$$

where the last inequality follows from standard properties of convex functions with Lipschitz continuous gradient, see e.g., [8, Lemma 3.4]. Taking the expectation and using the fact that $\mathbb{E}[\zeta\zeta^\top] = (1/d)I_{d \times d}$ we obtain

$$\mathbb{E}[(f(x + hr\zeta) - f(x - hr\zeta))^2 K^2(r)] \leq 3\kappa \left( \frac{\bar{L}^2 h^4}{2} + \frac{4h^2}{d} \|\nabla f(x)\|^2 \right). \tag{17}$$

To prove part (i) of the lemma, it is enough to combine (16), (17) and the inequality $\|\nabla f(x)\| \leq \bar{L}\|x - x^*\|$ that follows from the Lipschitz gradient assumption and the fact that $\nabla f(x^*) = 0$. Next, under the assumptions of part (ii) of the lemma we get analogously to (17) that

$$\big(f(x + hr\zeta) - f(x - hr\zeta)\big)^2 \leq 3\left( 2L^2 \|hr\zeta\|^4 + 4\langle \nabla f(x), hr\zeta \rangle^2 \right).$$

This yields inequality (17) with the only difference that $\bar{L}^2/2$ is replaced by $2L^2$. Together with (16), it implies the result. ∎

**Lemma A.1.** *Let $f$ be Lipschitz continuous with constant $G > 0$ in a Euclidean $h_t$-neighborhood of the set $\Theta$, and let Assumption 2.1 (i) hold. Let $\hat{g}_t$ and $x_t$ be defined by Algorithm 1. Then*

$$\mathbb{E}[\|\hat{g}_t\|^2 \,|\, x_t] \leq \kappa\left( C^* G^2 d + \frac{3d^2}{2h_t^2}\sigma^2 \right),$$

*where $C^* > 0$ is a numerical constant and $\kappa = \int K^2(u)du$.*

**Proof.** We have

$$
\begin{aligned}
\|\hat{g}\|^2 &= \frac{d^2}{4h^2} \|(f(x + hr\zeta) - f(x - hr\zeta) + \xi - \xi')\zeta K(r)\|^2 \\
&= \frac{d^2}{4h^2} (f(x + hr\zeta) - f(x - hr\zeta) + \xi - \xi')^2 K^2(r).
\end{aligned}
$$

Using the inequality $(a + b + c)^2 \leq 3(a^2 + b^2 + c^2)$ we get

$$\mathbb{E}[\|\hat{g}\|^2 \,|\, x] \leq \frac{3d^2}{4h^2} \left( \mathbb{E}[(f(x + hr\zeta) - f(x - hr\zeta))^2 K^2(r)] + 2\kappa\sigma^2 \right).$$

The lemma now follows by using [33, Lemma 10], which shows by a concentration argument that if $x \in \Theta$, $r \in [-1, 1]$ are fixed, $\zeta$ is uniformly distributed on the unit sphere and $f$ is Lipschitz continuous with constant $G > 0$ in a Euclidean $h$-neighborhood of the set $\Theta$, then

$$\mathbb{E}[(f(x + hr\zeta) - f(x - hr\zeta))^2] \leq c\frac{(hr)^2 G^2}{d},$$

where $c > 0$ is a numerical constant. ∎

**Lemma A.2.** *Let $f(\cdot)$ be a convex function on $\mathbb{R}^d$ and $h_t > 0$. Then the following holds.*

  *(i) Function $\hat{f}_t(\cdot)$ is convex on $\mathbb{R}^d$.*

*(ii)* $\hat{f}_t(x) \geq f(x)$ *for all* $x \in \mathbb{R}^d$.

*(iii)* *Function* $\hat{f}_t(\cdot)$ *is differentiable on* $\mathbb{R}^d$ *and for the conditional expectation given* $x_t$ *we have*

$$\mathbb{E}[\hat{g}_t|x_t] = \nabla \hat{f}_t(x_t).$$

**Proof.** Item (i) is straightforward. To prove item (ii), consider $g_t \in \partial f(x)$. Then,

$$\hat{f}_t(x) \geq \mathbb{E}[f(x) + h_t\langle g_t, \tilde{\zeta}\rangle] = f(x) + h_t\langle g_t, \mathbb{E}[\tilde{\zeta}]\rangle = f(x).$$

For item (iii) we refer to [25, pg. 350], or [16]. It is based on the fact that for any $x \in \mathbb{R}^d$ using Stokes formula we have

$$\nabla\hat{f}_t(x) = \frac{1}{V(B_d)h_t^d}\int_{\|v\|=h_t} f(x+v)\frac{v}{\|v\|}\mathrm{d}s_{h_t}(v) = \frac{d}{V(S_d)h_t}\int_{\|u\|=1} f(x+h_t u)u\,\mathrm{d}s_1(u)$$

$$= \frac{d}{V(S_d)h_t}\int_{\|u\|=1} f(x+h_t u)u\,\mathrm{d}s_1(u) = \mathbb{E}\Big[\frac{d}{h_t}f(x+h_t\zeta_t)\zeta_t\Big]$$

where $V(B_d)$ is the volume of the unit ball $B_d$, $\mathrm{d}s_r(\cdot)$ is the element of spherical surface of raduis $r$ in $\mathbb{R}^d$, and $V(S_d) = dV(B_d)$ is the surface area of the unit sphere in $\mathbb{R}^d$. Since $f(x + h_t\zeta_t)\zeta_t$ has the same distribution as $f(x - h_t\zeta_t)(-\zeta_t)$ we also get

$$\mathbb{E}\Big[\frac{d\big(f(x+h_t\zeta_t) - f(x-h_t\zeta_t)\big)\zeta_t}{2h_t}\Big] = \nabla\hat{f}_t(x).$$

∎

**Lemma A.3.** *If $f$ is $\alpha$-strongly convex then $\hat{f}_t$ is $\alpha$-strongly convex. If $f \in \mathcal{F}_2(L)$ then for any $x \in \mathbb{R}^d$ and $h_t > 0$ we have*

$$|\hat{f}_t(x) - f(x)| \leq Lh_t^2. \tag{18}$$

*and*

$$|\mathbb{E}f(x \pm h_t\zeta_t) - f(x)| \leq Lh_t^2. \tag{19}$$

**Proof.** Using the fact that $\mathbb{E}[\tilde{\zeta}] = 0$ we have

$$\big|\mathbb{E}\big[f(x+h_t\tilde{\zeta}) - f(x)\big]\big| = \big|\mathbb{E}\big[f(x+h_t\tilde{\zeta}) - f(x) - \langle\nabla f(x), h_t\tilde{\zeta}\rangle\big]\big| \leq Lh_t^2\mathbb{E}[\|\tilde{\zeta}\|^2] \leq Lh_t^2.$$

Thus, (18) follows. The proof of (19) is analogous. The $\alpha$-strong convexity of $\hat{f}_t$ is equivalent to the relation

$$\langle\nabla\hat{f}_t(x) - \nabla\hat{f}_t(x'), x - x'\rangle \geq \alpha\|x - x'\|^2, \quad \forall x, x' \in \mathbb{R}^d, \tag{20}$$

which is proved as follows:

$$\langle\nabla\hat{f}_t(x) - \nabla\hat{f}_t(x'), x - x'\rangle = \langle\mathbb{E}\big[\nabla f(x+h_t\tilde{\zeta}) - \nabla f(x'+h_t\tilde{\zeta})\big], x - x'\rangle \tag{21}$$

$$= \mathbb{E}\big[\langle\nabla f(x+h_t\tilde{\zeta}) - \nabla f(x'+h_t\tilde{\zeta}), (x+h_t\tilde{\zeta}) - (x'+h_t\tilde{\zeta})\rangle\big]$$

$$\geq \alpha\|x - x'\|^2, \quad \forall x, x' \in \mathbb{R}^d,$$

due to the $\alpha$-strong convexity of $f$. ∎

## B   Proofs

**Theorem 3.1.** (Upper Bound, Constrained Case.)   *Let $f \in \mathcal{F}_{\alpha,\beta}(L)$ with $\alpha, L > 0$ and $\beta \geq 2$. Let Assumptions 2.1 and 2.2 hold and let $\Theta$ be a convex compact subset of $\mathbb{R}^d$. Assume that $\max_{x\in\Theta}\|\nabla f(x)\| \leq G$. If $\sigma > 0$ then the cumulative regret of Algorithm 1 with*

$$h_t = \left(\frac{3\kappa\sigma^2}{2(\beta-1)(\kappa_\beta L)^2}\right)^{\frac{1}{2\beta}} t^{-\frac{1}{2\beta}}, \quad \eta_t = \frac{2}{\alpha t}, \quad t = 1, \ldots, T$$

*satisfies*

$$\forall x \in \Theta : \sum_{t=1}^{T} \mathbb{E}[f(x_t) - f(x)] \leq \frac{1}{\alpha} \left( d^2 \left( A_1 T^{1/\beta} + A_2 \right) + A_3 d \log T \right), \tag{4}$$

*where* $A_1 = 3\beta(\kappa\sigma^2)^{\frac{\beta-1}{\beta}}(\kappa_\beta L)^{\frac{2}{\beta}}$, $A_2 = \bar{c}\bar{L}^2(\sigma/L)^{\frac{2}{\beta}} + 9\kappa G^2/d$ *with constant* $\bar{c} > 0$ *depending only on* $\beta$, *and* $A_3 = 9\kappa G^2$. *The optimization error of averaged estimator* $\bar{x}_T = \frac{1}{T}\sum_{t=1}^{T} x_t$ *satisfies*

$$\mathbb{E}[f(\bar{x}_T) - f(x^*)] \leq \frac{1}{\alpha} \left( d^2 \left( \frac{A_1}{T^{\frac{\beta-1}{\beta}}} + \frac{A_2}{T} \right) + A_3 \frac{d \log T}{T} \right), \tag{5}$$

*where* $x^* = \arg\min_{x \in \Theta} f(x)$. *If* $\sigma = 0$, *then the cumulative regret and the optimization error of Algorithm 1 with any* $h_t$ *chosen small enough and* $\eta_t = \frac{2}{\alpha t}$ *satisfy the bounds* (4) *and* (5), *respectively, with* $A_1 = 0$, $A_2 = 9\kappa G^2/d$ *and* $A_3 = 10\kappa G^2$.

**Proof.** Fix an arbitrary $x \in \Theta$. By the definition of the algorithm, we have $\|x_{t+1} - x\|^2 \leq \|x_t - \eta_t \hat{g}_t - x\|^2$, which is equivalent to

$$\langle \hat{g}_t, x_t - x \rangle \leq \frac{\|x_t - x\|^2 - \|x_{t+1} - x\|^2}{2\eta_t} + \frac{\eta_t}{2}\|\hat{g}_t\|^2. \tag{22}$$

By the strong convexity assumption we have

$$f(x_t) - f(x) \leq \langle \nabla f(x_t), x_t - x \rangle - \frac{\alpha}{2}\|x_t - x\|^2. \tag{23}$$

Combining the last two displays and setting $a_t = \|x_t - x\|^2$ we obtain

$$
\begin{aligned}
\mathbb{E}[f(x_t) - f(x)\,|\,x_t] &\leq \|\mathbb{E}[\hat{g}_t\,|\,x_t] - \nabla f(x_t)\|\|x_t - x\| + \frac{1}{2\eta_t}\mathbb{E}[a_t - a_{t+1}\,|\,x_t] \\
&\quad + \frac{\eta_t}{2}\mathbb{E}[\|\hat{g}_t\|^2\,|\,x_t] - \frac{\alpha}{2}\mathbb{E}[a_t\,|\,x_t] \\
&\leq \kappa_\beta L d h_t^{\beta-1}\|x_t - x\| + \frac{1}{2\eta_t}\mathbb{E}[a_t - a_{t+1}\,|\,x_t] \\
&\quad + \frac{\eta_t}{2}\mathbb{E}[\|\hat{g}_t\|^2\,|\,x_t] - \frac{\alpha}{2}\mathbb{E}[a_t\,|\,x_t],
\end{aligned} \tag{24}
$$

where the second inequality follows from Lemma 2.3. As $2ab \leq a^2 + b^2$ we have

$$d h_t^{\beta-1}\|x_t - x\| \leq \frac{1}{2}\left( \frac{2\kappa_\beta L}{\alpha}d^2 h_t^{2(\beta-1)} + \frac{\alpha}{2\kappa_\beta L}\|x_t - x\|^2 \right). \tag{25}$$

We conclude, taking the expectations and letting $r_t = \mathbb{E}[a_t]$, that

$$\mathbb{E}[f(x_t) - f(x)] \leq \frac{r_t - r_{t+1}}{2\eta_t} - \frac{\alpha}{4}r_t + (\kappa_\beta L)^2 \frac{d^2}{\alpha} h_t^{2(\beta-1)} + \frac{\eta_t}{2}\mathbb{E}[\|\hat{g}_t\|^2] \tag{26}$$

Summing both sides over $t$ gives

$$\sum_{t=1}^{T} \mathbb{E}[f(x_t) - f(x)] \leq \frac{1}{2}\sum_{t=1}^{T} \left( \frac{r_t - r_{t+1}}{\eta_t} - \frac{\alpha}{2}r_t \right) + \sum_{t=1}^{T} \left( (\kappa_\beta L)^2 \frac{d^2}{\alpha} h_t^{2(\beta-1)} + \frac{\eta_t}{2}\mathbb{E}[\|\hat{g}_t\|^2] \right).$$

The first sum on the r.h.s. is smaller than 0 for our choice of $\eta_t = \frac{2}{\alpha t}$. Indeed,

$$\sum_{t=1}^{T} \left( \frac{r_t - r_{t+1}}{\eta_t} - \frac{\alpha}{2}r_t \right) \leq r_1\left( \frac{1}{\eta_1} - \frac{\alpha}{2} \right) + \sum_{t=2}^{T} r_t \left( \frac{1}{\eta_t} - \frac{1}{\eta_{t-1}} - \frac{\alpha}{2} \right) = 0.$$

From this remark and Lemma 2.4(ii) (where we use that Assumption 2.2 implies $f \in \mathcal{F}_2(\bar{L}/2)$) we obtain

$$
\begin{aligned}
\sum_{t=1}^{T} \mathbb{E}[f(x_t) - f(x)] &\leq \frac{1}{\alpha}\sum_{t=1}^{T} \left( (\kappa_\beta L)^2 d^2 h_t^{2(\beta-1)} + \frac{1}{t}\mathbb{E}[\|\hat{g}_t\|^2] \right) \tag{27} \\
&\leq \frac{1}{\alpha}\sum_{t=1}^{T} \left( (\kappa_\beta L)^2 d^2 h_t^{2(\beta-1)} + \frac{1}{t}\left[ 9\kappa\left( G^2 d + \frac{\bar{L}^2 d^2 h_t^2}{8} \right) + \frac{3\kappa d^2 \sigma^2}{2h_t^2} \right] \right) \\
&\leq \frac{d^2}{\alpha}\sum_{t=1}^{T} \left[ \left\{ (\kappa_\beta L)^2 h_t^{2(\beta-1)} + \frac{3}{2}\frac{\kappa\sigma^2}{h_t^2 t} \right\} + \frac{9\kappa\bar{L}^2 h_t^2}{8t} \right] + \frac{9\kappa G^2}{\alpha}d(\log T + 1). \tag{28}
\end{aligned}
$$

If $\sigma > 0$ then our choice of $h_t = \left(\frac{3\kappa\sigma^2}{2(\beta-1)(\kappa_\beta L)^2}\right)^{\frac{1}{2\beta}} t^{-\frac{1}{2\beta}}$ is the minimizer of the main term (in curly brackets in (28)). Plugging this $h_t$ in (28) and using the fact that $\sum_{t=1}^{T} t^{-1+1/\beta} \leq \beta T^{1/\beta}$ for $\beta \geq 2$ we get (4). Inequality (5) follows from (4) in view of the convexity of $f$. If $\sigma = 0$ the stochastic variability term in (28) disappears and one can choose $h_t$ as small as desired, in particular, such that the sum in (28) is smaller than $\frac{\kappa G^2}{\alpha} d \log T$. This yields the bounds for $\sigma = 0$. ∎

**Theorem 4.1.** *Let the assumptions of Theorem 3.1 be satisfied. Let $\sigma > 0$ and assume that $(\xi_t'')_{t=1}^{T}$ are independent random variables with $\mathbb{E}[\xi_t''] = 0$ and $\mathbb{E}[\xi_t''] \leq \sigma^2$ for $t = 1, \dots, T$. If $f$ attains its minimum at point $x^* \in \Theta$, then*

$$\mathbb{E}|\hat{M} - f(x^*)| \leq \frac{\sigma}{T^{\frac{1}{2}}} + \frac{1}{\alpha}\left(d^2\left(\frac{A_1}{T^{\frac{\beta-1}{\beta}}} + \frac{A_2}{T}\right) + A_3\frac{d\log T}{T}\right). \tag{9}$$

**Proof.** We have

$$\mathbb{E}|\hat{M} - f(x^*)| \leq \mathbb{E}\left|\frac{1}{T}\sum_{t=1}^{T}\xi_t''\right| + \mathbb{E}\left|\frac{1}{T}\sum_{t=1}^{T}(f(x_t) - f(x^*))\right|$$

$$= \mathbb{E}\left|\frac{1}{T}\sum_{t=1}^{T}\xi_t''\right| + \frac{1}{T}\sum_{t=1}^{T}\mathbb{E}[f(x_t) - f(x^*)]$$

$$\leq \frac{\sigma}{T^{\frac{1}{2}}} + \frac{1}{T}\sum_{t=1}^{T}\mathbb{E}[f(x_t) - f(x^*)]$$

and the theorem follows by using (4). ∎

**Theorem 3.2.** (Upper Bounds, Unconstrained Case.) *Let $f \in \mathcal{F}_{\alpha,\beta}(L)$ with $\alpha, L > 0$ and $\beta \geq 2$. Let Assumptions 2.1 and 2.2 hold. Assume also that $\alpha > \sqrt{C_* d/T}$, where $C_* > 72\kappa\bar{L}^2$. Let $x_t$'s be the updates of Algorithm 1 with $\Theta = \mathbb{R}^d$, $h_t$ and $\eta_t$ as in (6) and a non-random $x_1 \in \mathbb{R}^d$. Then the estimator defined by (7) satisfies*

$$\mathbb{E}[f(\bar{x}_{T_0,T}) - f(x^*)] \leq C\kappa\bar{L}^2\frac{d}{\alpha T}\|x_1 - x^*\|^2 + C\frac{d^2}{\alpha}\left((\kappa_\beta L)^2 + \kappa(\bar{L}^2 + \sigma^2)\right)T^{-\frac{\beta-1}{\beta}} \tag{8}$$

*where $C > 0$ is a constant depending only on $\beta$ and $x^* = \arg\min_{x\in\mathbb{R}^d} f(x)$.*

**Proof.** We start as in the proof of Theorem 3.1 to get (24). Then, using the strong convexity of $f$ and the fact that $x^*$ is the minimizer of $f$ we get analogously to (25) that

$$dh_t^{\beta-1}\|x_t - x^*\| \leq \frac{1}{2}\left(\frac{2\kappa_\beta L}{\alpha}d^2 h_t^{2(\beta-1)} + \frac{\alpha}{2\kappa_\beta L}\|x_t - x^*\|^2\right) \leq \frac{\kappa_\beta L}{\alpha}d^2 h_t^{2(\beta-1)} + \frac{f(x_t) - f(x^*)}{2\kappa_\beta L}.$$

Combining the last display and (24), using Lemma 2.4 and letting $r_t = \mathbb{E}[\|x_t - x^*\|^2]$ we get

$$\mathbb{E}[f(x_t) - f(x^*)] \leq \frac{r_t - r_{t+1}}{\eta_t} - \alpha r_t + 2(\kappa_\beta L)^2\frac{d^2}{\alpha}h_t^{2(\beta-1)} + \kappa\eta_t\left[9\bar{L}^2\left(dr_t + \frac{d^2 h_t^2}{8}\right) + \frac{3d^2\sigma^2}{2h_t^2}\right]. \tag{29}$$

For $t = 1, \dots, T_0$, since $h_t = T^{-\frac{1}{2\beta}}$ and $\eta_t = (\alpha T)^{-1}$ we have the following consequence of (29)

$$r_{t+1} \leq r_t\left(1 - \frac{1}{T} + \frac{9\kappa\bar{L}^2}{(\alpha T)^2}d\right) + b_T \leq r_t\left(1 + \frac{9\kappa\bar{L}^2}{(\alpha T)^2}d\right) + b_T \tag{30}$$

where

$$b_T = \frac{d^2}{\alpha^2 T}\left(2(\kappa_\beta L)^2 T^{-\frac{\beta-1}{\beta}} + \frac{9}{8}\kappa\bar{L}^2 T^{-\frac{\beta+1}{\beta}} + \frac{3}{2}\kappa\sigma^2 T^{-\frac{\beta-1}{\beta}}\right) \leq$$

$$\leq \frac{d^2}{\alpha^2 T}\left(2(\kappa_\beta L)^2 + \frac{9}{8}\kappa\bar{L}^2 + \frac{3}{2}\kappa\sigma^2\right)T^{-\frac{\beta-1}{\beta}}. \tag{31}$$

Letting $C_3 = 9\kappa \bar{L}^2$, inequality (30) is of the form $r_{t+1} \leq r_t q + b_T$, with $q = (1 + \frac{C_3 d}{(\alpha T)^2})$. Then

$$r_{T_0+1} \leq r_1 q^{T_0} + b_T \sum_{j=1}^{T_0-1} q^j \leq r_1 q^{T_0} + b_T \frac{q^{T_0}}{q-1} \leq \left( r_1 + \frac{(\alpha T)^2}{C_3 d} b_T \right) q^{T_0}.$$

Now, assuming

$$T_0 = \left\lfloor \frac{4 C_3 d}{\alpha^2} \right\rfloor \tag{32}$$

we obtain

$$
\begin{aligned}
q^{T_0} &= \exp\left[ T_0 \log\left( 1 + \frac{C_3 d}{(\alpha T)^2} \right) \right] \\
&\leq \exp\left[ \frac{4 C_3 d}{\alpha^2} \log\left( 1 + \frac{C_3 d}{(\alpha T)^2} \right) \right] \\
&\leq \exp\left( \frac{4 C_3^2 d^2}{\alpha^4 T^2} \right) \leq \exp\left( \frac{4 C_3^2}{C_*^2} \right) =: C_4
\end{aligned}
$$

where in the last inequality we have used the assumption that, for $C_* > 0$ large enough,

$$\alpha > \sqrt{\frac{C_* d}{T}}. \tag{33}$$

As we shall see, this also guarantees that $T_0 < T$. In conclusion, we obtain

$$
\begin{aligned}
r_{T_0+1} &\leq C_4\left( r_1 + \frac{(\alpha T)^2}{C_3 d} b_T \right) \\
&\leq C_4\left( r_1 + \frac{(\alpha T)^2}{C_3 d} \frac{d^2}{\alpha^2 T} \left( 2(\kappa_\beta L)^2 + \frac{9}{8}\kappa \bar{L}^2 + \frac{3}{2}\kappa \sigma^2 \right) T^{-\frac{\beta-1}{\beta}} \right) \\
&= C_4\left( r_1 + \frac{d}{C_3} \left( 2(\kappa_\beta L)^2 + \frac{9}{8}\kappa \bar{L}^2 + \frac{3}{2}\kappa \sigma^2 \right) T^{\frac{1}{\beta}} \right). \tag{34}
\end{aligned}
$$

We now go back to inequality (29). Recalling the definition of $\bar{x}_{T_0,T}$ and the fact that $h_t = t^{-\frac{1}{2\beta}}$ and $\eta_t = \frac{2}{\alpha t}$ for $t \in \{T_0+1, \dots T\}$, we deduce from (29) that

$$
\begin{aligned}
(T - T_0)\mathbb{E}[f(\bar{x}_{T_0,T}) - f(x^*)] &\leq \sum_{t=T_0+1}^{T} (r_t - r_{t+1})\frac{\alpha t}{2} - \alpha r_t + 18\kappa \frac{\bar{L}^2}{\alpha t} d r_t \\
&\quad + \frac{d^2}{\alpha} \sum_{t=T_0+1}^{T} \left( 2(\kappa_\beta L)^2 t^{-\frac{\beta-1}{\beta}} + \frac{9}{4}\kappa \bar{L}^2 t^{-\frac{\beta+1}{\beta}} + 3\kappa\sigma^2 t^{-\frac{\beta-1}{\beta}} \right).
\end{aligned}
$$

Since $9\kappa \bar{L}^2 = C_3$ condition (32) implies that $\frac{18\kappa \bar{L}^2}{\alpha t} d \leq \frac{\alpha}{2}$ for $t \geq T_0 + 1$. Thus

$$(T - T_0)\mathbb{E}[f(\bar{x}_{T_0,T}) - f(x^*)] \leq \frac{\alpha}{2} \sum_{t=T_0+1}^{T} \left[ (r_t - r_{t+1})t - r_t \right] + U_T,$$

where

$$U_T = \frac{d^2}{\alpha}\left( 2(\kappa_\beta L)^2 + \frac{9}{4}\kappa \bar{L}^2 + 3\kappa\sigma^2 \right) \sum_{t=T_0}^{T} t^{-\frac{\beta-1}{\beta}} \leq \frac{d^2}{\alpha}\left( 2(\kappa_\beta L)^2 + \frac{9}{4}\kappa \bar{L}^2 + 3\kappa\sigma^2 \right) \beta T^{\frac{1}{\beta}}.$$

On the other hand

$$\sum_{t=T_0+1}^{T} \left[ (r_t - r_{t+1})t - r_t \right] \leq r_{T_0+1}(T_0 + 1 - 1) + \sum_{t=T_0+2}^{T} r_t(t - (t-1) - 1) = T_0 r_{T_0+1}.$$

Using inequality (34) and condition (32) we get

$$
\begin{aligned}
\frac{\alpha T_0}{2} r_{T_0+1} &\leq \frac{2C_3 C_4 d}{\alpha} \left( r_1 + \frac{d}{C_3} \left( 2(\kappa_\beta L)^2 + \frac{9}{8}\kappa\bar{L}^2 + \frac{3}{2}\kappa\sigma^2 \right) T^{\frac{1}{\beta}} \right) \\
&= 2C_4 \left( 9\kappa\bar{L}^2 \frac{d}{\alpha} r_1 + \frac{d^2}{\alpha} \left( 2(\kappa_\beta L)^2 + \frac{9}{8}\kappa\bar{L}^2 + \frac{3}{2}\kappa\sigma^2 \right) T^{\frac{1}{\beta}} \right).
\end{aligned}
$$

These bounds imply

$$
(T-T_0)\mathbb{E}[f(\bar{x}_{T_0,T}) - f(x^*)] \leq 18C_4\kappa\bar{L}^2 \frac{d}{\alpha} r_1 + (2C_4+\beta)\frac{d^2}{\alpha}\left(2(\kappa_\beta L)^2 + \frac{9}{4}\kappa\bar{L}^2 + 3\kappa\sigma^2\right)T^{\frac{1}{\beta}}.
$$

Since $C_* > 8C_3 = 72\kappa\bar{L}^2$ it follows from (32) and (33) that $T \geq 2T_0$. Thus

$$
\mathbb{E}[f(\bar{x}_{T_0,T}) - f(x^*)] \leq 36C_4\kappa\bar{L}^2 \frac{d}{\alpha T} r_1 + \left(4C_4+2\beta\right)\frac{d^2}{\alpha}\left(2(\kappa_\beta L)^2 + \frac{9}{4}\kappa\bar{L}^2 + 3\kappa\sigma^2\right)T^{-\frac{\beta-1}{\beta}}.
$$

∎

**Theorem 5.1.** *Let $f \in \mathcal{F}_{\alpha,2}(L)$ with $\alpha, L > 0$. Let Assumption 2.1 hold and let $\Theta$ be a convex compact subset of $\mathbb{R}^d$. Assume that $\max_{x \in \Theta} \|\nabla f(x)\| \leq G$. If $\sigma > 0$ then for Algorithm 1 with $\hat{g}_t$ defined in (10) and parameters $h_t = \left(\frac{3d^2\sigma^2}{4L\alpha t + 9L^2 d^2}\right)^{1/4}$ and $\eta_t = \frac{1}{\alpha t}$ we have*

$$
\forall x \in \Theta: \quad \mathbb{E}\sum_{t=1}^{T}\left(f(x_t) - f(x)\right) \leq \min\left(GBT, 2\sqrt{3L}\sigma\frac{d}{\sqrt{\alpha}}\sqrt{T} + A_4\frac{d^2}{\alpha}\log T\right), \quad (11)
$$

*where $B$ is the Euclidean diameter of $\Theta$ and $A_4 = 6.5L\sigma + 22G^2/d$. Moreover, if $x^* = \arg\min_{x \in \Theta} f(x)$ the optimization error of averaged estimator $\bar{x}_T = \frac{1}{T}\sum_{t=1}^{T} x_t$ is bounded as*

$$
\mathbb{E}[f(\bar{x}_T) - f(x^*)] \leq \min\left(GB, 2\sqrt{3L}\sigma\frac{d}{\sqrt{\alpha T}} + A_4\frac{d^2}{\alpha}\frac{\log T}{T}\right). \quad (12)
$$

*Finally, if $\sigma = 0$, then the cumulative regret of Algorithm 1 with any $h_t$ chosen small enough and $\eta_t = \frac{1}{\alpha t}$ and the optimization error of its averaged version are of the order $\frac{d^2}{\alpha}\log T$ and $\frac{d^2}{\alpha}\frac{\log T}{T}$, respectively.*

**Proof.** Fix $x \in \Theta$. Due to the $\alpha$-strong convexity of $\hat{f}_t$ (cf. Lemma A.3) we have

$$
\hat{f}_t(x_t) - \hat{f}_t(x) \leq \langle \nabla\hat{f}_t(x_t), x_t - x \rangle - \frac{\alpha}{2}\|x_t - x\|^2.
$$

Using (18) and Lemma A.2(ii) we obtain

$$
f(x_t) - f(x) \leq Lh_t^2 + \langle \nabla\hat{f}_t(x_t), x_t - x \rangle - \frac{\alpha}{2}\|x_t - x\|^2.
$$

Using this property and exploiting inequality (22) we find, with an argument similar to the proof of Theorem 3.1, that

$$
\forall x \in \Theta: \quad \mathbb{E}\left[f(x_t) - f(x)\right] \leq Lh_t^2 + \frac{r_t - r_{t+1}}{2\eta_t} - \frac{\alpha}{2}r_t + \frac{\eta_t}{2}\mathbb{E}[\|\hat{g}_t\|^2]. \quad (35)
$$

By assumption, $\eta_t = \frac{1}{\alpha t}$. Summing up from $t = 1$ to $T$ and reasoning again analogously to the proof of Theorem 3.1 we obtain

$$
\forall x \in \Theta: \quad \mathbb{E}\sum_{t=1}^{T}\left(f(x_t) - f(x)\right) \leq \sum_{t=1}^{T}\left(Lh_t^2 + \frac{1}{2\alpha t}\mathbb{E}[\|\hat{g}_t\|^2]\right). \quad (36)
$$

Now, inspection of the proof of Lemma 2.4 shows that it remains valid with $\kappa = 1$ when $K(\cdot) \equiv 1$ in Algorithm 1. This yields

$$
\mathbb{E}[\|\hat{g}_t\|^2] \leq 9\left(G^2 d + \frac{L^2 d^2 h_t^2}{2}\right) + \frac{3d^2\sigma^2}{2h_t^2}.
$$

Thus,

$$\forall x \in \Theta: \qquad \mathbb{E}\sum_{t=1}^{T}\big(f(x_t) - f(x)\big) \leq \sum_{t=1}^{T}\Big[\Big(L + \frac{9L^2 d^2}{4\alpha t}\Big)h_t^2 + \frac{3d^2\sigma^2}{4h_t^2\alpha t} + \frac{9G^2 d}{2\alpha t}\Big]. \qquad (37)$$

The chosen value $h_t = \Big(\frac{3d^2\sigma^2}{4L\alpha t + 9L^2 d^2}\Big)^{1/4}$ minimizes the r.h.s. and yields

$$\forall x \in \Theta: \ \mathbb{E}\sum_{t=1}^{T}\big(f(x_t) - f(x)\big) \leq \frac{3}{2}\sum_{t=1}^{T}\frac{d^2\sigma^2}{\alpha t}\Big(\frac{4L\alpha t + 9L^2 d^2}{3d^2\sigma^2}\Big)^{1/2} + \frac{9G^2}{2}\frac{d}{\alpha}(1 + \log T)$$

$$\leq \sum_{t=1}^{T}\sqrt{3}\Big[\frac{d\sigma\sqrt{L}}{\sqrt{\alpha t}} + \frac{3Ld^2\sigma}{2\alpha t}\Big] + 9G^2\frac{d}{\alpha}(1 + \log T)$$

$$\leq 2\sqrt{3L}\sigma\frac{d}{\sqrt{\alpha}}\sqrt{T} + \Big(\frac{3\sqrt{3}}{2}\sigma L + \frac{9G^2}{d}\Big)\frac{d^2}{\alpha}(1 + \log T).$$

As $1 + \log T \leq ((\log 2)^{-1} + 1)\log T$ for any $T \geq 2$, we obtain (11). On the other hand, we have the straightforward bound

$$\forall x \in \Theta: \qquad \mathbb{E}\sum_{t=1}^{T}\big(f(x_t) - f(x)\big) \leq GBT. \qquad (38)$$

The remaining part of the proof follows the same lines as in Theorem 3.1. ∎

**Theorem 6.1.** *Let $\Theta = \{x \in \mathbb{R}^d : \|x\| \leq 1\}$. For $\alpha, L > 0, \beta \geq 2$, let $\mathcal{F}'_{\alpha,\beta}$ denote the set of functions $f$ that attain their minimum over $\mathbb{R}^d$ in $\Theta$ and belong to $\mathcal{F}_{\alpha,\beta}(L) \cap \{f : \max_{x \in \Theta}\|\nabla f(x)\| \leq G\}$, where $G > 2\alpha$. Then for any strategy in the class $\Pi_T$ we have*

$$\sup_{f \in \mathcal{F}'_{\alpha,\beta}} \mathbb{E}\big[f(z_T) - \min_x f(x)\big] \geq C \min\Big(\max(\alpha, T^{-1/2+1/\beta}), \frac{d}{\sqrt{T}}, \frac{d}{\alpha}T^{-\frac{\beta-1}{\beta}}\Big), \qquad (14)$$

*and*

$$\sup_{f \in \mathcal{F}'_{\alpha,\beta}} \mathbb{E}\big[\|z_T - x^*(f)\|^2\big] \geq C \min\Big(1, \frac{d}{T^{\frac{1}{\beta}}}, \frac{d}{\alpha^2}T^{-\frac{\beta-1}{\beta}}\Big), \qquad (15)$$

*where $C > 0$ is a constant that does not depend of $T, d$, and $\alpha$, and $x^*(f)$ is the minimizer of $f$ on $\Theta$.*

**Proof.** We use the fact that $\sup_{f \in \mathcal{F}'_{\alpha,\beta}}$ is bigger than the maximum over a finite family of functions in $\mathcal{F}'_{\alpha,\beta}$. We choose this finite family in a way that its members cannot be distinguished from each other with positive probability but are separated enough from each other to guarantee that the maximal optimization error for this family is of the order of the desired lower bound.

We first assume that $\alpha \geq T^{-1/2+1/\beta}$.

Let $\eta_0 : \mathbb{R} \to \mathbb{R}$ be an infinitely many times differentiable function such that

$$\eta_0(x) = \begin{cases} = 1 & \text{if } |x| \leq 1/4, \\ \in (0,1) & \text{if } 1/4 < |x| < 1, \\ = 0 & \text{if } |x| \geq 1. \end{cases}$$

Set $\eta(x) = \int_{-\infty}^{x}\eta_0(\tau)d\tau$. Let $\Omega = \{-1,1\}^d$ be the set of binary sequences of length $d$. Consider the finite set of functions $f_\omega : \mathbb{R}^d \to \mathbb{R}, \omega \in \Omega$, defined as follows:

$$f_\omega(u) = \alpha(1+\delta)\|u\|^2/2 + \sum_{i=1}^{d}\omega_i r h^\beta \eta(u_i h^{-1}), \qquad u = (u_1, \ldots, u_d),$$

where $\omega_i \in \{-1,1\}, h = \min\big((\alpha^2/d)^{\frac{1}{2(\beta-1)}}, T^{-\frac{1}{2\beta}}\big)$ and $r > 0, \delta > 0$ are fixed numbers that will be chosen small enough.

Let us prove that $f_\omega \in \mathcal{F}'_{\alpha,\beta}$ for $r > 0$ and $\delta > 0$ small enough. It is straightforward to check that if $r$ is small enough the functions $f_\omega$ are $\alpha$-strongly convex and belong to $\mathcal{F}_\beta(L)$.

Next, the components of the gradient $\nabla f_\omega$ have the form

$$(\nabla f_\omega(u))_i = \alpha(1+\delta)u_i + \omega_i r h^{\beta-1}\eta_0(u_i h^{-1}).$$

Thus,

$$\|\nabla f_\omega(u)\|^2 \leq 2\alpha^2(1+\delta)^2 \|u\|^2 + 2r^2\alpha^2$$

and the last expression can be rendered smaller than $G^2$ uniformly in $u \in \Theta$ by the choice of $\delta$ and $r$ small enough since $G^2 > 4\alpha^2$.

Finally, we check that the minimizers of functions $f_\omega$ belong to $\Theta$. Notice that we can choose $r$ small enough to have $\alpha^{-1}(1+\delta)^{-1}rh^{\beta-2} < 1/4$ and that under this condition the equation $\nabla f_\omega(x) = 0$ has the solution

$$x_\omega^* = (x^*(\omega_1), \ldots, x^*(\omega_d)),$$

where $x^*(\omega_i) = -\omega_i \alpha^{-1}(1+\delta)^{-1}rh^{\beta-1}$. Using the definition of $h$ we obtain

$$\|x_\omega^*\| \leq d^{1/2}\alpha^{-1}(1+\delta)^{-1}rh^{\beta-1} \leq d^{1/2}\alpha^{-1}(1+\delta)^{-1}r(\alpha^2/d)^{1/2} \leq (1+\delta)^{-1}r < 1$$

for $r > 0$ small enough, which means that $x_\omega^*$ belongs to the interior of $\Theta$.

Combining all the above remarks we conclude that the family of functions $\{f_\omega, \omega \in \Omega\}$ is a subset of $\mathcal{F}'_{\alpha,\beta}$ for $r > 0$ and $\delta > 0$ small enough.

For any fixed $\omega \in \Omega$, we denote by $\mathbf{P}_{\omega,T}$ the probability measure corresponding to the joint distribution of $(z_1, y_1, \ldots, y_T)$ where $y_t = f_\omega(z_t) + \xi_t$ with independent identically distributed $\xi_t$'s such that (13) holds and $z_t$'s chosen by a sequential strategy in $\Pi_T$. We have

$$d\mathbf{P}_{\omega,T}(z_1, y_1, \ldots, y_T) = dF\big(y_1 - f_\omega(z_1)\big)\prod_{i=2}^T dF\Big(y_i - f_\omega\big(\Phi_i(z_1, y_1, \ldots, y_{i-1})\big)\Big).$$

Without loss of generality, we omit here the dependence of $\Phi_i$ on $z_2, \ldots, z_{i-1}$ since $z_i, i \geq 2$, is a Borel function of $z_1, y_1, \ldots, y_{i-1}$. Let $\mathbf{E}_{\omega,T}$ denote the expectation w.r.t. $\mathbf{P}_{\omega,T}$.

Consider the statistic

$$\hat\omega \in \arg\min_{\omega \in \Omega} \|z_T - x_\omega^*\|.$$

Since $\|x_{\hat\omega}^* - x_\omega^*\| \leq \|z_T - x_\omega^*\| + \|z_T - x_{\hat\omega}^*\| \leq 2\|z_T - x_\omega^*\|$ for all $\omega \in \Omega$ we obtain

$$\begin{aligned}
\mathbf{E}_{\omega,T}\big[\|z_T - x_\omega^*\|^2\big] &\geq \frac{1}{4}\mathbf{E}_{\omega,T}\big[\|x_\omega^* - x_{\hat\omega}^*\|^2\big] \\
&= \alpha^{-2}r^2 h^{2\beta-2}\mathbf{E}_{\omega,T}\rho(\hat\omega, \omega),
\end{aligned}$$

where $\rho(\hat\omega, \omega) = \sum_{i=1}^d \mathbb{I}(\hat\omega_i \neq \omega_i)$ is the Hamming distance between $\hat\omega$ and $\omega$. Taking the maximum over $\Omega$ and then the minimum over all statistics $\hat\omega$ with values in $\Omega$ we obtain

$$\max_{\omega \in \Omega}\mathbf{E}_{\omega,T}\big[\|z_T - x_\omega^*\|^2\big] \geq \alpha^{-2}r^2 h^{2\beta-2}\inf_{\hat\omega}\max_{\omega \in \Omega}\mathbf{E}_\omega\rho(\hat\omega, \omega).$$

By [34, Theorem 2.12], if for some $\gamma > 0$ and all $\omega, \omega' \in \Omega$ such that $\rho(\omega, \omega') = 1$ we have $KL(\mathbf{P}_{\omega,T}, \mathbf{P}_{\omega',T}) \leq \gamma$, where $KL(\cdot, \cdot)$ denotes the Kullback-Leibler divergence, then

$$\inf_{\hat\omega}\max_{\omega \in \Omega}\mathbf{E}_{\omega,T}\rho(\hat\omega, \omega) \geq \frac{d}{4}\exp(-\gamma).$$

Now for all $\omega, \omega' \in \Omega$ such that $\rho(\omega, \omega') = 1$ we have

$$KL(\mathbf{P}_{\omega,T}, \mathbf{P}_{\omega',T}) = \int \log\Big(\frac{d\mathbf{P}_{\omega,T}}{d\mathbf{P}_{\omega',T}}\Big) d\mathbf{P}_{\omega,T}$$

$$= \int \Big[ \log\Big(\frac{dF(y_1 - f_\omega(z_1))}{dF(y_1 - f_{\omega'}(z_1))}\Big) +$$

$$+ \sum_{i=2}^{T} \log\Big(\frac{dF\big(y_i - f_\omega\big(\Phi_i(z_1, y_1^{i-1})\big)\big)}{dF\big(y_i - f_{\omega'}\big(\Phi_i(z_1, y_1^{i-1})\big)\big)}\Big)\Big]$$

$$dF\big(y_1 - f_\omega(z_1)\big) \prod_{i=2}^{T} dF\Big(y_i - f_\omega\big(\Phi_i(z_1, y_1^{i-1})\big)\Big)$$

$$\leq T I_0 \max_{u \in \mathbb{R}} |f_\omega(u) - f_{\omega'}(u)|^2 = I_0 r^2 \eta^2(1),$$

where the last inequality is granted if $r < v_0/\eta(1)$ due to (13). Assuming in addition that $r$ satisfies $r^2 \leq (\log 2)/\big(I_0 \eta^2(1)\big)$ we obtain $KL(\mathbf{P}_{\omega,T}, \mathbf{P}_{\omega',T}) \leq \log 2$. Therefore, we have proved that if $\alpha \geq T^{-1/2+1/\beta}$ then there exist $r > 0$ and $\delta > 0$ small enough such that

$$\max_{\omega \in \Omega} \mathbf{E}_{\omega,T}\big[\|z_T - x_\omega^*\|^2\big] \geq \frac{1}{8} d\alpha^{-2} r^2 h^{2\beta-2} = \frac{r^2}{8} \min\Big(1, \frac{d}{\alpha^2} T^{-\frac{\beta-1}{\beta}}\Big). \tag{39}$$

This implies (15) for $\alpha \geq T^{-1/2+1/\beta}$. In particular, if $\alpha = \alpha_0 := T^{-1/2+1/\beta}$ the bound (39) is of the order $\min\Big(1, d T^{-\frac{1}{\beta}}\Big)$. Then for $0 < \alpha < \alpha_0$ we also have the bound of this order since the classes $\mathcal{F}'_{\alpha,\beta}$ are nested: $\mathcal{F}'_{\alpha_0,\beta} \subset \mathcal{F}'_{\alpha,\beta}$. This completes the proof of (15).

We now prove (14). From (39) and $\alpha$-strong convexity of $f$ we get that, for $\alpha \geq T^{-1/2+1/\beta}$,

$$\max_{\omega \in \Omega} \mathbf{E}_{\omega,T}\big[f(z_T) - f(x_\omega^*)\big] \geq \frac{r^2}{16} \min\Big(\alpha, \frac{d}{\alpha} T^{-\frac{\beta-1}{\beta}}\Big). \tag{40}$$

This implies (14) in the zone $\alpha \geq T^{-1/2+1/\beta}$ since for such $\alpha$ we have

$$\min\Big(\alpha, \frac{d}{\alpha} T^{-\frac{\beta-1}{\beta}}\Big) = \min\Big(\max(\alpha, T^{-1/2+1/\beta}), \frac{d}{\sqrt{T}}, \frac{d}{\alpha} T^{-\frac{\beta-1}{\beta}}\Big).$$

On the other hand,

$$\min\Big(\alpha_0, \frac{d}{\alpha_0} T^{-\frac{\beta-1}{\beta}}\Big) = \min\Big(T^{-1/2+1/\beta}, \frac{d}{\sqrt{T}}\Big),$$

and the same lower bound holds for $0 < \alpha < \alpha_0$ by the nestedness argument that we used to prove (15) in the zone $0 < \alpha < \alpha_0$. Thus, (14) follows.

∎

## C   Comments on [3]

In this section we comment on issues with some claims in the paper of Bach and Perchet [3], which presents a number of valuable results and provides a motivation for our work. We wish to clarify such issues for the sake of understanding, as otherwise a comparison to the results presented here would be misleading.

Bach and Perchet [3] introduce Algorithm 1 in the current form and provide upper bounds for its optimisation error and online regret when $f \in \mathcal{F}_\beta(L)$ with integer $\beta$. The setting where $f$ is strongly convex is considered in Propositions 4,6-8 and 9 of that paper. Propositions 4, 6,9 give the rates decaying in $T$ not faster than $T^{-\frac{\beta-1}{\beta+1}}$, which is slower than the optimal rate $T^{-\frac{\beta-1}{\beta}}$. Proposition 8 dealing with asymptotic results is problematic. It is stated as bounds on $\|x_N - x^*\|$ but the authors presumably mean bounds on $\mathbb{E}\|x_N - x^*\|^2$. The proof relies on the last inequality of Lemma 2 in [3], where factor $d$ is missing. The right-hand side of this inequality should be of the order $d\delta^{\beta-1}$ and not $\delta^{\beta-1}$ (this is analogous to our Lemma 2.3). This leads to too optimistic dependency of the bounds in Proposition 8 on the dimension $d$. The same issue arises in Proposition 5 (the second line of its proof uses a bound on the norm of $\zeta_n$ with missing factor $d$). A dependency of the bound on the initial value of the algorithm is missing in the part of Proposition 8 entitled "unconstrained optimization of strongly convex mappings". This remark also concerns Proposition 7.

# D Additional results

In this appendix, we provide refined versions of Theorems 3.1 and 5.1. First we state a non-asymptotic version of Chung's lemma [11, Lemma 1]. It allows us to obtain in Theorem D.2 upper bounds for $\mathbb{E}\{\|x_t - x^*\|^2\}$, where $x_t$ is generated by a constrained version of Algorithm 1 (i.e., with compact $\Theta$) under the assumptions of Theorems 3.1 and 5.1. By using this result and considering averaging from $\lfloor T/2 \rfloor + 1$ to $T$ rather than from 1 to $T$, in Theorems D.3 and D.4 we provide finer upper bounds for the optimization error than in Theorems 3.1 and 5.1. The refinement consists in the fact that we get rid of the logarithmic factors appearing in (5) and (12). Finally, in Theorem D.5 we show that the term $\frac{d^2}{\alpha} \log T$ in the bound on the cumulative regret in Theorem 5.1 can be improved to $\frac{d}{\alpha} \log T$ under a slightly more restrictive assumption (we assume that the norm $\|\nabla f\|$ is uniformly bounded by $G$ on a large enough Euclidean neighborhood of $\Theta$ rather than only on $\Theta$).

**Lemma D.1.** *Let $\{b_t\}$ be a sequence of real numbers such that for all integers $t \geq 2$,*

$$b_{t+1} < \left(1 - \frac{1}{t}\right) b_t + \sum_{i=1}^{N} \frac{a_i}{t^{p_i+1}}, \tag{41}$$

*where $0 < p_i < 1$ and $a_i \geq 0$ for $1 \leq i \leq N$. Then for $t \geq 2$ we have*

$$b_t < \frac{2b_2}{t} + \sum_{i=1}^{N} \frac{a_i}{(1-p_i)t^{p_i}}. \tag{42}$$

**Proof.** For any fixed $t > 0$ the convexity of the mapping $u \mapsto g(u) = (t + u)^{-p}$ implies that $g(1) - g(0) \geq g'(0)$, i.e.,

$$\frac{1}{t^p} - \frac{1}{(t+1)^p} \leq \frac{p}{t^{p+1}}.$$

Thus,

$$\frac{a_i}{t^{p+1}} \leq \frac{a_i}{1-p} \left( \frac{1}{(t+1)^p} - \left(1 - \frac{1}{t}\right) \frac{1}{t^p} \right). \tag{43}$$

Using (41), and (43) and rearranging terms we get

$$b_{t+1} - \sum_{i=1}^{N} \frac{a_i}{(1-p_i)(t+1)^{p_i}} \leq \left(1 - \frac{1}{t}\right) \left[ b_t - \sum_{i=1}^{N} \frac{a_i}{(1-p_i)t^{p_i}} \right].$$

Letting $\tau_t = b_t - \sum_{i=1}^{N} \frac{a_i}{(1-p_i)t^{p_i}}$ we have $\tau_{t+1} \leq (1 - \frac{1}{t})\tau_t$. Now, if $\tau_2 \leq 0$ then $\tau_t \leq 0$ for any $t \geq 2$ and thus (42) holds. Otherwise, if $\tau_2 > 0$ then for $t \geq 3$ we have

$$\tau_t \leq \tau_2 \prod_{i=2}^{t-1} \left(1 - \frac{1}{i}\right) \leq \frac{2\tau_2}{t} \leq \frac{2b_2}{t},$$

where we have used the inequalities $\sum_{i=2}^{t-1} \log\left(1 - \frac{1}{i}\right) \leq -\sum_{i=2}^{t-1} \frac{1}{i} \leq -\log(t-1) \leq \log(2/t)$. Thus, (42) holds in this case as well. ∎

**Theorem D.2.** *Let $f \in \mathcal{F}_{\alpha,\beta}(L)$ with $\beta \geq 2$, $\alpha, L > 0$, $\sigma > 0$, and let Assumption 2.1 hold. Consider Algorithm 1 where $\Theta$ is a convex compact subset of $\mathbb{R}^d$ and assume that $\max_{x \in \Theta} \|\nabla f(x)\| \leq G$.*

*(i) If Assumption 2.2 holds, $h_t = \left(\frac{3\kappa\sigma^2}{2(\beta-1)(\kappa_\beta L)^2}\right)^{\frac{1}{2\beta}} t^{-\frac{1}{2\beta}}$ and $\eta_t = \frac{2}{\alpha t}$ then for $t \geq 1$ we have*

$$\mathbb{E}\big[\|x_t - x^*\|^2\big] < \frac{2G^2}{\alpha^2 t} + A_5 \frac{d^2}{\alpha^2} t^{-\frac{\beta-1}{\beta}} \tag{44}$$

*where $x^* = \arg\min_{x \in \Theta} f(x)$ and $A_5 > 0$ is a constant that does not depend on $d, \alpha, t$.*

*(ii) If $\beta = 2$, $h_t = \left(\frac{3d^2\sigma^2}{4L\alpha t + 9L^2 d^2}\right)^{1/4}$ and $\eta_t = \frac{1}{\alpha t}$ then for $t \geq 1$ we have that*

$$\mathbb{E}\big[\|x_t - x^*\|^2\big] < \frac{2G^2}{\alpha^2 t} + A_6 \frac{d}{\alpha^{\frac{3}{2}} t^{\frac{1}{2}}} + A_7 \frac{d^2}{\alpha^2 t}, \tag{45}$$

*where $A_6, A_7 > 0$ are constants that do not depend on $d, \alpha, t$.*

**Proof.** Let $r_t = \mathbb{E}\|x_t - x^*\|^2$. To prove the theorem, we will show that under the assumptions of the theorem $\{r_t\}$ satisfies (41) with suitable $a_i$ and $p_i$, and then use Lemma D.1.

We start by noticing that, in view of the $\alpha$-strong convexity of $f$ and the fact that $f$ is Lipschitz continuous with constant $G$ in $\Theta$ for any $t \geq 1$ we have

$$\|x_t - x^*\|^2 \leq \frac{G^2}{\alpha^2}. \tag{46}$$

Thus, (44) and (45) hold for $t = 1$ and it suffices to prove the theorem for $t \geq 2$. The definition of Algorithm 1 gives that, for $t \geq 1$,

$$\|x_{t+1} - x^*\|^2 \leq \|x_t - x^*\|^2 - 2\eta_t \langle \hat{g}_t, x_t - x^* \rangle + \eta_t^2 \|\hat{g}_t\|^2 .$$

Taking conditional expectation of both sides of this inequality given $x_t$ we obtain

$$\mathbb{E}[\|x_{t+1} - x^*\|^2 \,|x_t] \leq \|x_t - x^*\|^2 - 2\eta_t \langle \mathbb{E}[\hat{g}_t|x_t], x_t - x^* \rangle + \eta_t^2 \mathbb{E}[\|\hat{g}_t\|^2 \,|x_t].$$

Using this inequality and Lemmas 2.3 and 2.4(ii) we find

$$\mathbb{E}[\|x_{t+1} - x^*\|^2 \,|x_t] \leq \|x_t - x^*\|^2 - 2\eta_t\alpha \|x_t - x^*\|^2 + 2\eta_t\kappa_\beta L d h_t^{\beta-1}\|x_t - x^*\| +$$
$$+\eta_t^2 \left[\left(9\kappa\left(G^2 d + \frac{L^2 d^2 h_t^2}{2}\right) + \frac{3\kappa d^2\sigma^2}{2h_t^2}\right)\right]. \tag{47}$$

On the other hand, for $\lambda > 0$, we have

$$d h_t^{\beta-1} \|x_t - x^*\| \leq \frac{1}{2}\left(\frac{\kappa_\beta L}{\alpha\lambda} d^2 h_t^{2(\beta-1)} + \frac{\alpha\lambda}{\kappa_\beta L}\|x_t - x^*\|^2\right). \tag{48}$$

Combining (48) and (47) we get

$$\mathbb{E}[\|x_{t+1} - x^*\|^2 \,|x_t] \leq (1 - (2-\lambda)\eta_t\alpha)\|x_t - x^*\|^2 + \frac{(\kappa_\beta L)^2}{\alpha\lambda}\eta_t d^2 h_t^{2(\beta-1)} +$$
$$+\eta_t^2 \left[\left(9\kappa\left(G^2 d + \frac{L^2 d^2 h_t^2}{2}\right) + \frac{3\kappa d^2\sigma^2}{2h_t^2}\right)\right]. \tag{49}$$

Substituting $h_t = \left(\frac{3\kappa\sigma^2}{2(\beta-1)(\kappa_\beta L)^2}\right)^{\frac{1}{2\beta}} t^{-\frac{1}{2\beta}}$, $\eta_t = \frac{2}{\alpha t}$, $\lambda = \frac{3}{2}$ in (49), and taking the expectation over $x_t$ we obtain

$$r_{t+1} \leq \left(1 - \frac{1}{t}\right)r_t + \frac{4(\kappa_\beta L)^2}{3\alpha^2}d^2\left(\frac{3\kappa\sigma^2}{2(\beta-1)(\kappa_\beta L)^2}\right)^{\frac{\beta-1}{\beta}} t^{-\frac{2\beta-1}{\beta}} +$$
$$+\frac{18\kappa L^2 d^2}{\alpha^2}\left(\frac{3\kappa\sigma^2}{2(\beta-1)(\kappa_\beta L)^2}\right)^{\frac{1}{\beta}} t^{-\frac{2\beta+1}{\beta}} + \frac{36\kappa}{\alpha^2 t^2}G^2 d +$$
$$+\frac{6\kappa d^2\sigma^2}{\alpha^2}\left(\frac{3\kappa\sigma^2}{2(\beta-1)(\kappa_\beta L)^2}\right)^{-\frac{1}{\beta}} t^{-\frac{2\beta-1}{\beta}}.$$

Thus, we have

$$r_{t+1} < \left(1 - \frac{1}{t}\right)r_t + C\frac{d^2}{\alpha^2}t^{-\frac{2\beta-1}{\beta}},$$

where

$$C = \frac{4(\kappa_\beta L)^2}{3}\left(\frac{3\kappa\sigma^2}{2(\beta-1)(\kappa_\beta L)^2}\right)^{\frac{\beta-1}{\beta}} + 18\kappa L^2\left(\frac{3\kappa\sigma^2}{2(\beta-1)(\kappa_\beta L)^2}\right)^{\frac{1}{\beta}} +$$
$$+\frac{36\kappa}{d}G^2 + 6\kappa\sigma^2\left(\frac{3\kappa\sigma^2}{2(\beta-1)(\kappa_\beta L)^2}\right)^{-\frac{1}{\beta}}.$$

This is a particular instance of (41). Therefore, we can apply Lemma D.1, which yields that, for all $t \geq 2$,

$$r_t < \frac{2G^2}{\alpha^2 t} + \beta C\frac{d^2}{\alpha^2}t^{-\frac{\beta-1}{\beta}}.$$

Thus, (44) follows.

We now prove (45). Since $\beta = 2$, using Lemmas 2.3, 2.4(ii), and A.3 we obtain

$$\mathbb{E}[\|x_{t+1} - x^*\|^2 \,|x_t] \le (1 - \eta_t \alpha)\, \|x_t - x^*\|^2 + 2\eta_t L h_t^2 + \eta_t^2 \left[\left(9\left(G^2 d + \frac{L^2 d^2 h_t^2}{2}\right) + \frac{3d^2\sigma^2}{2h_t^2}\right)\right].$$

Setting here $h_t = \left(\frac{3d^2\sigma^2}{4L\alpha t + 9L^2 d^2}\right)^{1/4}$, $\eta_t = \frac{1}{\alpha t}$, and taking the expectation over $x_t$ we get

$$
\begin{aligned}
r_{t+1} &\le \left(1 - \frac{1}{t}\right)r_t + \left(\frac{(4L\alpha t + 9L^2 d^2)^{1/2}}{\alpha^2}\right)\frac{\sqrt{3}d\sigma}{t^2} + \frac{9G^2 d}{\alpha^2 t^2} \\
&\le \left(1 - \frac{1}{t}\right)r_t + A_6' \frac{d}{\alpha^{\frac{3}{2}} t^{\frac{3}{2}}} + A_7' \frac{d^2}{\alpha^2 t^2},
\end{aligned}
$$

where $A_6' = 2\sqrt{3L}\sigma$ and $A_7' = 3\sqrt{3L}\sigma + \frac{9G^2}{d}$. Applying Lemma D.1 for $t \ge 2$ we get

$$r_t < \frac{2G^2}{\alpha^2 t} + 2A_6' \frac{d}{\alpha^{\frac{3}{2}} t^{\frac{1}{2}}} + 2A_7' \frac{d^2}{\alpha^2 t}.$$

∎

Consider the estimator

$$\hat{x}_T = \frac{1}{T - \lfloor T/2 \rfloor} \sum_{t=\lfloor T/2 \rfloor + 1}^{T} x_t. \tag{50}$$

The following two theorems provide bounds on the optimization error of this estimator.

**Theorem D.3.** *Let $f \in \mathcal{F}_{\alpha,\beta}(L)$ with $\beta \ge 2$, $\alpha, L > 0$, $\sigma > 0$, and let Assumptions 2.1 and 2.2 hold. Consider Algorithm 1 where $\Theta$ is a convex compact subset of $\mathbb{R}^d$ and assume that $\max_{x \in \Theta}\|\nabla f(x)\| \le G$. If $h_t = \left(\frac{3\kappa\sigma^2}{2(\beta-1)(\kappa_\beta L)^2}\right)^{\frac{1}{2\beta}} t^{-\frac{1}{2\beta}}$ and $\eta_t = \frac{2}{\alpha t}$ then the optimization error of the estimator (50) satisfies*

$$\mathbb{E}[f(\hat{x}_T) - f(x^*)] \le \min\left(GB, \frac{1}{\alpha}\left(d^2\left(\frac{A_1'}{T^{\frac{\beta-1}{\beta}}} + \frac{A_2'}{T}\right) + \frac{A_3' d}{T}\right)\right),$$

*where $x^* = \arg\min_{x \in \Theta} f(x)$. Here $A_1'$, $A_2'$ and $A_3'$ are positive constants that do not depend on $d, \alpha, T$, and $B$ is the Euclidean diameter of $\Theta$.*

**Proof.** With the same steps as in the proof of Theorem 3.1 (see (28)) but taking now the sum over $t = \lfloor T/2 \rfloor + 1, \ldots, T$ rather than over $t = 1, \ldots, T$ we obtain

$$
\begin{aligned}
\sum_{t=\lfloor T/2 \rfloor + 1}^{T} \mathbb{E}[f(x_t) - f(x^*)] &\le r_{\lfloor T/2 \rfloor + 1} \frac{\lfloor T/2 \rfloor \alpha}{2} + \frac{1}{\alpha} \sum_{t=\lfloor T/2 \rfloor + 1}^{T} \left((\kappa_\beta L)^2 d^2 h_t^{2(\beta-1)} + \right. \\
&\quad \left. + \frac{1}{t}\left[9\kappa\left(G^2 d + \frac{\bar{L}^2 d^2 h_t^2}{8}\right) + \frac{3\kappa d^2 \sigma^2}{2h_t^2}\right]\right) \\
&\le r_{\lfloor T/2 \rfloor + 1} \frac{\lfloor T/2 \rfloor \alpha}{2} + \frac{9\kappa G^2 d}{\alpha} \sum_{t=\lfloor T/2 \rfloor + 1}^{T} \frac{1}{t} \\
&\quad + \frac{1}{\alpha} \sum_{t=1}^{T} \left((\kappa_\beta L)^2 d^2 h_t^{2(\beta-1)} + \frac{\bar{L}^2 d^2 h_t^2}{8t} + \frac{3\kappa d^2 \sigma^2}{2h_t^2 t}\right).
\end{aligned}
$$

For the last sum here, we use exactly the same bound as in the proof of Theorem 3.1. Moreover, it follows from Theorem D.2 that

$$r_{\lfloor T/2 \rfloor + 1} < \frac{4G^2}{\alpha^2 T} + A_5' \frac{d^2}{\alpha^2} T^{-\frac{\beta-1}{\beta}},$$

where $A_5' = 2^{(\beta-1)/\beta} A_5$. Combining these remarks and using the fact that $\sum_{t=\lfloor T/2 \rfloor+1}^{T} \frac{1}{t} \leq \log(T/\lfloor T/2 \rfloor) \leq 2$ for all $T \geq 2$ (recall that we assume $T \geq 2$ throughout the paper), as well as the the convexity of $f$ we get

$$\mathbb{E}[f(\hat{x}_T) - f(x^*)] \quad \leq \quad \frac{1}{\alpha}\left(d^2\left(\frac{A_1'}{T^{\frac{\beta-1}{\beta}}} + \frac{A_2'}{T}\right) + \frac{A_3'd}{T}\right),$$

where $A_1' = 2A_1 + \frac{A_5'}{2}$, $A_2' = 2\bar{c}\bar{L}^2(\sigma/L)^{\frac{2}{\beta}}$ with constant $\bar{c}$ as in Theorem 3.1 and $A_3' = 2G^2(18\kappa + 1/d)$. On the other hand we have the straightforward bound

$$\mathbb{E}[f(\hat{x}_T) - f(x^*)] \leq GB.$$

∎

**Theorem D.4.** *Let $f \in \mathcal{F}_{\alpha,2}(L)$ with $\alpha, L > 0$, $\sigma > 0$, and let Assumption 2.1 hold. Consider the version of Algorithm 1 as in Theorem 5.1 where $\Theta$ is a convex compact subset of $\mathbb{R}^d$ and assume that $\max_{x \in \Theta} \|\nabla f(x)\| \leq G$. If $h_t = \left(\frac{3d^2\sigma^2}{4L\alpha t + 9L^2 d^2}\right)^{1/4}$ and $\eta_t = \frac{1}{\alpha t}$ then the optimization error of the estimator (50) satisfies*

$$\mathbb{E}[f(\hat{x}_T) - f(x^*)] \leq \min\left(GB, A_8\frac{d}{\sqrt{\alpha T}} + A_9\frac{d^2}{\alpha T}\right), \tag{51}$$

*where $x^* = \arg\min_{x \in \Theta} f(x)$. Here $A_8$ and $A_9$ are positive constants that do not depend on $d, \alpha, T$, and $B$ is the Euclidean diameter of $\Theta$.*

**Proof.** Arguing as in the proof of Theorem 5.1 but taking the sum over $\lfloor T/2 \rfloor + 1, \ldots, T$ rather than over $1, \ldots, T$ we obtain

$$\sum_{t=\lfloor T/2 \rfloor+1}^{T} \mathbb{E}[f(x_t) - f(x^*)] \leq r_{\lfloor T/2 \rfloor+1}\frac{\lfloor T/2 \rfloor \alpha}{2} + \sum_{t=\lfloor T/2 \rfloor+1}^{T}\left[\left(L + \frac{9L^2d^2}{4\alpha t}\right)h_t^2 + \frac{3d^2\sigma^2}{4h_t^2\alpha t} + \frac{9G^2 d}{2\alpha t}\right]$$

$$\leq r_{\lfloor T/2 \rfloor+1}\frac{\lfloor T/2 \rfloor \alpha}{2} + \sum_{t=\lfloor T/2 \rfloor+1}^{T}\left[\sqrt{3}\frac{d\sigma\sqrt{L}}{\sqrt{\alpha t}} + \frac{3\sqrt{3}Ld^2\sigma}{2\alpha t} + \frac{9G^2 d}{2\alpha t}\right]$$

$$\leq r_{\lfloor T/2 \rfloor+1}\frac{\lfloor T/2 \rfloor \alpha}{2} + 2\sqrt{3L}\sigma\frac{d}{\sqrt{\alpha}}\sqrt{T} + \frac{3d}{2\alpha}(\sqrt{3}Ld\sigma + 3G^2)\sum_{t=\lfloor T/2 \rfloor+1}^{T}\frac{1}{t}$$

$$\leq r_{\lfloor T/2 \rfloor+1}\frac{\lfloor T/2 \rfloor \alpha}{2} + 2\sqrt{3L}\sigma\frac{d}{\sqrt{\alpha}}\sqrt{T} + \frac{3d}{\alpha}(\sqrt{3}Ld\sigma + 3G^2),$$

where we have used the inequality $\sum_{t=\lfloor T/2 \rfloor+1}^{T} \frac{1}{t} \leq \log(T/\lfloor T/2 \rfloor) \leq 2$ for all $T \geq 2$ (recall that we assume $T \geq 2$ throughout the paper). It follows from Theorem D.2, that

$$r_{\lfloor T/2 \rfloor+1} < \frac{4G^2}{\alpha^2 T} + \sqrt{2}A_6\frac{d}{\alpha^{\frac{3}{2}}T^{\frac{1}{2}}} + 2A_7\frac{d^2}{\alpha^2 T}.$$

Combining the last two displays yields

$$\sum_{t=\lfloor T/2 \rfloor+1}^{T} \mathbb{E}[f(x_t) - f(x^*)] \leq \frac{G^2}{\alpha} + A_6\frac{d}{2\sqrt{2}\sqrt{\alpha}}\sqrt{T} + A_7\frac{d^2}{2\alpha} + 2\sqrt{3L}\sigma\frac{d}{\sqrt{\alpha}}\sqrt{T} + \frac{3d}{\alpha}(\sqrt{3}Ld\sigma + 3G^2).$$

From this inequality, using the fact that $f$ is a convex function, we obtain

$$\mathbb{E}[f(\hat{x}_T) - f(x^*)] \leq A_8\frac{d}{\sqrt{\alpha T}} + A_9\frac{d^2}{\alpha T},$$

where $A_8 = \frac{A_6}{\sqrt{2}} + 4\sqrt{3L}\sigma$ and $A_9 = A_7 + 2(3\sqrt{3}L\sigma + (9d+1)G^2/d^2)$. ∎

**Theorem D.5.** *Let $f \in \mathcal{F}_{\alpha,2}(L)$ with $\alpha, L > 0$, and let Assumption 2.1 hold. Consider the version of Algorithm 1 as in Theorem 5.1 where $\Theta$ is a convex compact subset of $\mathbb{R}^d$, and $h_t = \left(\frac{3d^2\sigma^2}{4L\alpha t}\right)^{\frac{1}{4}}$, $\eta_t = \frac{1}{\alpha t}$. If $f$ is Lipschitz continuous with Lipschitz constant $G$ on the Euclidean $h_1$-neighborhood of $\Theta$, then for $\sigma > 0$ we have the following bound for the cumulative regret:*

$$\forall x \in \Theta : \sum_{t=1}^{T} \mathbb{E}[f(x_t) - f(x)] \leq \min\left(GBT, 2\sqrt{3L}\sigma \frac{d}{\sqrt{\alpha}}\sqrt{T} + \frac{C^*G^2}{2}\frac{d}{\alpha}(1 + \log T)\right), \quad (52)$$

*where $B$ is the Euclidean diameter of $\Theta$.*

*If $\sigma = 0$, then the cumulative regret for any $h_t$ chosen small enough and $\eta_t = \frac{1}{\alpha t}$ satisfies*

$$\forall x \in \Theta : \sum_{t=1}^{T} \mathbb{E}[f(x_t) - f(x)] \leq \min\left(GBT, C^*G^2\frac{d}{\alpha}(1 + \log T)\right)$$

**Proof.** The argument is analogous to the proof of Theorem 5.1. The difference is only in the bound on $\mathbb{E}[\|\hat{g}_t\|^2]$. To evaluate this term, we now use Lemma A.1 (noticing that when $K(\cdot) \equiv 1$ this lemma is satisfied with $\kappa = 1$). This yields

$$\forall x \in \Theta : \qquad \mathbb{E}\sum_{t=1}^{T}\left(f(x_t) - f(x)\right) \leq \sum_{t=1}^{T}\left[Lh_t^2 + \frac{1}{2\alpha t}\left(C^*G^2 d + \frac{3d^2\sigma^2}{2h_t^2}\right)\right]. \quad (53)$$

The chosen value $h_t = \left(\frac{3d^2\sigma^2}{4L\alpha t}\right)^{\frac{1}{4}}$ minimizes the r.h.s. and together with (38) yields (52). The remaining part of the proof follows the same lines as in Theorem 3.1. ∎