[Reviews · NeurIPS 2020]

Review 1

Summary and Contributions: This paper studies the closely related problems of continuous stochastic bandits and zero-order stochastic optimization. The overall goal is to minimize an unknown strongly convex function by exploring its function values sequentially, where querying a point returns a noisy estimate of the true value of the function at that specific location. These problems are fundamental to many problems in bandits where the derivatives of the functions are too expensive to compute (or don't have a closed form expression), and so algorithms need to be developed that exploit higher order smoothness conditions without resorting to using gradient information. The goal then is to develop algorithms which utilize only zero-th order information on the function which has low optimization complexity (additive deviation of the point versus the true minimizer) and regret (cumulative additive error). This paper develops upper bounds on the optimization complexity and regret for gradient-like algorithms along with nearly-matching lower bounds. These bounds show explicit dependence on the number of iterations, convexity parameter, Holder smoothness parameter, etc, unlike prior bounds which just deals with dependence on the number of iterations. They also utilize a kernel to help exploit the higher order smoothness properties when interpolating the gradient. The overall set-up is as follows: there is an unknown alpha-strongly convex function d-dimensional function f which satisfies the \beta-Holder smoothness condition. The authors restrict to the case when \beta >= 2 so that they can impose an additional assumption that the function has Lipschitz smooth gradients. At every iteration the agent can query the point at several locations (taken to be at most three in the case of this algorithm), where they observe a noisy observation of the function value where the second moment of the noise is unbounded. (Note that this doesn't mean that the noise is zero-mean, which is somewhat interesting and is handled due to the convexity assumption and randomized algorithm). The goal then is threefold. First, minimize optimization complexity (namely the expected deviation of f(x_hat) - f(x^\star)) where x_hat is the estimate of the minimizer. Second, the cumulative regret (namely the additive deviation of f(x_t) - f(x^\star) where x_t is the point chosen by the algorithm). And lastly, an estimate for f(x^\star) where the goal is to minimize the expected deviation of M_hat and f(x^\star) where M_hat is the estimate. At a high level - the authors provide upper bounds on each of these quantities. They consider both the constrained and unconstrained setting, and the case under no noise or adversarial noise. Lastly, they show a lower bound on the optimization error which matches their bound up to a linear term on the dimension d. The overall algorithm is a simple (projected) gradient-like procedure with the addition of a kernel which is used to exploit the smoothness. At every iteration, the current estimate x_t for the minimizer of the function is taken to build two perturbations taken (roughly) uniformly inside of the h_t - sphere where h_t dictates how far away from the current iterate x_t the perturbation is chosen. Afterwards the agent querires the function value at these two points, uses the observed function value to compute an estimate of the gradient (which is not necessarily unbiased due to the noise model), and then takes a gradient step in the direction of the gradient (where the kernel is used to smooth the gradient). The authors then provide bounds for the three goals discussed before. The first of which on the cumulative regret, which scales linearly in the dimension of the space, and provide explicit dependence on the relevant parameters \beta and \alpha. In order to estimate the minimizer of the function, the authors suggest using the average of the iterates (or the delayed average of iterates) and provide a similar bound. Lastly, in order to estimate the function value of the minimizer (as the function is unknown you can't just take f(estimate of the minimizer)) they propose averaging the observed iterates of the function value and provide an upper bound on this setting. Moreover, for the case when \beta = 2 they offer improved bounds which scale linearly on the dimension instead of quadratically.

Strengths: This paper provides improved bounds for zero-th order stochastic gradient descent for the case where the function is alpha-strongly convex and \beta Holder smoothe. While the algorithm is based on existing literature, (with derivative free stochastic optimization dating back to Nemirovski), they provide novel bounds with explicit dependence on the relevant parameters d, \alpha, and \beta, unlike prior work which only shows explicit dependence on the number of iterations. In particular, the bounds that they show include: - upper bound on the optimization complexity - upper bound on the regret - upper bound on estimation error of the minimal value - a minimax lower bound on the optimization complexity for the case of independent noise (which matches their provided upper bound up to a factor of d) Moreover, the ability to obtain bounds on estimating the minimizer of the function at a rate of 1 / \sqrt{T} , which is the same statistical complexity one would obtain from just querying the minimizer over independent noise is both surprising and an interesting result. I believe that these contributions are of interest to the NeurIPS community, namely due to the importance of understanding zero-th order optimization in various machine learning tasks. This work serves at a starting point for understanding the impact of higher-order smoothness conditions on zero-th order optimization.

Weaknesses: The authors provide no discussion on the broader impacts. Clearly zero-th order optimization is very important for various stochastic bandit tasks, as many problems (e.g. optimizing complex chemical systems) have function values which are more readily computable than their gradient information. However, the authors provide no motivation, background / interesting problems to help put their work in perspective. Especially the additional assumption of strong convexity (which allows them to get guarantees which scale linearly with respect to the dimension instead of exponentially) is a BIG additional assumption in comparison to bandit literature, and there was no discussion / motivating problems as to when situations like this might arise. In addition, some of the related work section mostly concerns the optimization perspective, and ignores a large amount of related work considering exploiting smoothness in the stochastic contextual bandit literature. More discussion on this point is in the related work section of the review.

Correctness: The technical contributions of the papers appear correct, subject to clarity of presentation (see question 5/8 below). However, several of the proofs are very technical and inside of the appendix adding a brief summary or discussion of the result before stating them would help with clarity. Moreover, there are instances in the proof where theorems from prior work are utilized with no explicit discussion or clarity as to what the result is or where and how it fits in the proof. This made some of the proofs difficult to follow. Lastly in several cases the authors state that the proofs extend to the case when the function f_t is allowed to depend on the iteration number instead of a fixed unknown function. Adding more discussion on this in the appendix, and potentially state the more general bound under this assumption would help clear those points up.

Clarity: While the paper is readable and well-written, there is certainly room for improvements in the clarity of the paper. Overall the paper read more as a technical report, and lacked any form of story or over-arching clarity that helped illustrate how novel their technical contributions were. In particular, commenting on the beginning about using kernels to smooth the gradient estimates, overall additional proof technique, and discussion and motivation to the problem would help. As the algorithm is an existing one, explaining the key theoretical insights will help motivate the paper more. Some overall comments are below: - Including the related work section at the end of the paper makes sense, especially as the comparisons are quite technical. However, it would be useful to include some comment at the beginning with the punchline to summarize the intuition behind the analysis in comparison to related work (e.g. our algorithms achieve explicit dependence on \alpha and \beta where prior work does not, arising because in the proof we did ___) - A discussion section on how the proof is different from what was done before would help, either because of a more careful analysis, something specific to the kernels, etc. - The introduction states that there is a "suitably chosen kernel K that allows us to take advantage of higher order smoothness of f". However, the content of the paper never explicitly discusses how the kernel allows them to obtain explicit dependence on the parameters in their bounds. They state that prior work considers a similar algorithm without a kernel, but again never describe how the kernel helps them in their algorithm. Moreover, the bounds are presented with a term \kappa_\beta which depends on the choice of kernel. Adding in explicit values of this parameter for various kernels which satisfy the assumption (e.g. legendre polynomials as discussed) would potentially help make this dependence more clear. - The authors state that for the case when \beta = 2, the kernel is redundant. Clarifying this point would help show why the kernels are needed for the gradient estimate for the case when \beta > 2. - The lower bound presented in Theorem 6.1 has a condition on the cumulative distribution function on the noise. Adding some discussion on this assumption rather than just providing an example would help clarify this result. - For the unconstrained case the learning rate for the algorithm is taken as two different values depending on the iteration number, but the need for the different cases was never discussed. - Several of the theorem statements impose additional assumptions which were never discussed. As an example, in Theorem 3.2 the strong convexity parameter is taken to be sufficiently large. - The definition of cumulative regret was never explicitly written (while it was explicit from the theorem statement), it would help to define it separately.

Relation to Prior Work: The authors provide a very detailed related work to summarize the differences in their approach to existing zero-th order optimization based approaches. In particular, most of the prior work considers settings where the noise is deterministic or Gaussian, in which they consider a general setting by allowing for adversarial noises. Moreover, they provide explicit dependence on the bounds in terms of the strongly convex parameter alpha and the dimension d, whereas prior work just showed bounds with an unspecified constant proportional to the number of iterations. However, as the contributions are technical, highlighting the key proof difference will help clarify their result against prior work. On the counter-side, exploiting smoothness in the bandit literature (without the convexity assumption) has been studied before, e.g. "Smooth Contextual Bandits: Bridging the Parametric and Non-differentiable Regret Regimes" by Yichun Hu, Nathan Kallus, Xiaojie Mao 2019. "Nonparametric bandits with covariates". Philippe Rigollet and Assaf Zeevi, 2010 "The Multi-Armed Bandit Problem with Covariates". Vianney Perchet and Philippe Rigollet. 2013 Adding more discussion on the related work in stochastic bandits would help motivate and position this work in the broader context of stochastic multi-armed bandits (and especially by differentiating by adding the additional strong-convexity asumption which results in bounds which are not exponential in the dimension).

Reproducibility: Yes

Additional Feedback: Please note that the following comments are taken with the main paper from the main submission, and the appendix taken from the supplementary. - The discussion in line 180 is an interesting point, that the algorithm is able to obtain matching mini-max bounds of the form 1 / \sqrt{T} estimate for the maximizer under additive noise even for the case when the true minimizer of the function is unknown. - The discussion before Theorem 5.1 helped make the result more readable and interpretable, and adding a brief discussion akin to this before each result would help with the clarity of presentation. - In line 95 you express various assumptions on the kernels that are used. Can you give examples of the related \kappa_\beta parameters for the kernels you state satisfy the assumptions (e.g. the weighted legendre polynomials)? - Each of the algorithms requires tuning parameters under the assumption that \alpha, \beta, lipschitz parameter, etc are all known in advanced. Are you able to show asymptotic guarantees in the case when these values are unknown? - In line 430 it seems as though you dropped the factor from multiplying by K^2(r) which gives the \kappa term in the front. - Prior to stating Lemma A.2 it would be helpful to redefine the function \hat{f}_t - For Appendix Section C it might be helpful to summarize some of the results presented in the paper [3] that you are referring to for readers who are unfamiliar with the work.


Review 2

Summary and Contributions: This paper studies the problem of zeroth order optimization of a strongly convex function, a problem already studied by Bach and Perchet. They analyze the algorithm of Bach and Perchet under additional assumptions, namely high-order smoothness (using a Hölder-like condition to quantify the smoothness) and derive bounds on the cumulative regret and optimization error in this setting. They also prove minimax lower bounds on this problem.

Strengths: The claims are all proved and the proofs seem technically correct. I appreciated to see that all constants are explicit in the statements of the theorems, even if this does not improve the readability of the paper. I also appreciated to see sketches of the proofs of the main theorems in the main text.

Weaknesses: I am unsure of the significance of this contribution. This paper highly builds on the previous algorithm of Bach and Perchet. The contribution of this work is the analysis of this algorithm under the high-order smoothness assumption, which does not seem especially interesting to me. On top of that, I am under the impression that the proofs are derived in the setting where all the parameters are known (strong convexity constant, smoothness parameters, sigma, etc) which does not seem realistic. I would have been interested in seeing numerical experiments. How can this algorithm be implemented in practice ? And how do you ensure that its implementation achieves the same bounds, without knowing all parameters ?

Correctness: Yes. In order to strengthen the claims that their analysis outperforms previous ones, it would have been interesting to add experiments.

Clarity: The paper is globally well-written despite some typos (line 83, "estimated" for example). However I did not feel at ease with the introduction. The authors do not really motivate their work and begin very quickly with technicalities and explanations about the algorithm in the first page. I think that the related work section could have been put in the beginning of the paper.

Relation to Prior Work: Except from its place at the end of the paper, I found the relative work section clear and well-written. The authors discuss clearly their differences with previous contributions. Appendix C is also relevant in this direction.

Reproducibility: No

Additional Feedback: A direction for improvement could be the implementation of the algorithm in a practical setting, or a discussion on the fact that all parameters have to be known to derive the bounds of the theorems. How could you achieve these bounds without knowing the parameters ? ### AFTER REBUTTAL ### I have read the rebuttal and the answers from the authors. They did not convince me of the significance of their work, which is still interesting from a mathematical point of view. I will therefore stick with my initial score.


Review 3

Summary and Contributions: The authors study the zeroth-other convex optimization problem and study a previous algorithm of Bach and Perchet under additional assumptions that the target function is beta-Holder smooth and alpha-strongly convex. The authors prove upper bounds on the regret when the only assumption on the noise is that the variance is bounded (though the algorithm needs to be turned with knowledge of this variance). The authors also provide a guarantee for estimation of f(x^*) for a modified algorithm that uses a third query to estimate the function value. The authors provide stronger convergence guarantees in the beta=2 case (exploiting the connection to the surrogate function), as well as lower bounds for arbitrary beta>=2.

Strengths: Baring my being ignorant of large part of this literature, the technical contributions seem strong. In particular, the algorithms analyzed are simple, intuitive, and computationally trivial. Pointers to improvements from the literature (such as the log(T) improvement taking a tail average would yield) are appreciated. Presenting a lower bound really strengthens the upper bounds in the paper and provides a more holistic picture of the algorithm, and the paper seems to provide a satisfactory addition to our understanding on how smoothness affects rates. Section 7 contextualizes the results very well. Given the fundamental nature of zeroth order approximation, I believe the results in this paper would be very relevant to the community.

Weaknesses: Sometimes the presentation is dense: a table, for example, would be a more efficient way to compare the derived rates with past results. There are a few discoveries I wish the authors would discuss a bit more, including: - the generalization to "adversarial noise;" e.g. explain why this generalization is plausible. - showing the bias-variance decomposition explicitly, at least one, would be nice - can you explain why the kernel is redundant when beta=2 (line 204)? - Since the claimed lower bound is novel, can you explain what is new about the construction?

Correctness: The proofs seem reasonable, but I didn't check them.

Clarity: Generally yes, but see some suggestions in 'weaknesses' above.'

Relation to Prior Work: Extensively.

Reproducibility: Yes

Additional Feedback: Why does def 2.1 need to constrain zeta and r? They are chosen by the learner. # post rebuttal: My questions were mostly addressed, though I still think def 2.1 is unnatural. I will keep my score.


Review 4

Summary and Contributions: The authors focus on the problem of zero-order optimization of a strongly convex function, in order to find the minimizer of the function via a sequential approach. They focus on the impact of higher-order smoothness on both the optimization error as well we on the cumulative regret. They consider a randomized approximation of the projected gradient descent algorithm, where the gradient is estimated via two function evaluations and a smoothing kernel. Several theoretical results are derived under different settings. Their results imply that the zero-order algorithm is nearly optimal in terms of sample complexity and problem parameters. They also provide an estimator of the minimum function value which achieves sharp oracle behavior.

Strengths: EDIT: I thank the authors for their response, which have addressed my questions. I leave my evaluation unchanged. ---------------------------------------------------------------------------------------------- The paper is very well written and is a pleasure to read. The main strengths are as follows: 1. Very clear problem statement and clean highlights of the contributions 2. The theoretical proofs seem sound (Unfortunately, I was not able to go through the detailed proofs, but the proof sketch seems correct and clearly highlights the steps needed to prove the claims. 3. The paper is highly novel as it improves on previous literature and considers settings that are much more general than before. Their result on the lower bound is very novel. 4. This is of high relevance to the NeurIPS community

Weaknesses: I didn't find any strong weakness in the paper but it might be good to add some empirical results on some functions to visually see how these bounds behave in practice.

Correctness: The sketch of the proof seem correct. The paper did not have any empirical evidence.

Clarity: It is very well written and is easy to follow. Few minor suggestions. 1. For the claim in line 180 and 181 can the authors add a reference? 2. Type in line 206: x -> x_t in defining y_t and y_t'

Relation to Prior Work: The related works section is excellent and clearly highlights how this work differs from previous contributions.

Reproducibility: Yes

Additional Feedback:

[Author Response · NeurIPS 2020]

We thank all reviewers for their valuable comments, which will be taken into account in the revision of the paper.

**R3, R5:** *Adding empirical results.* We agree that an extensive numerical experiment would be of interest. We did not
develop it in this paper as it is already dense in theoretical results. The current format allows maybe for adding a small
numerical example, which cannot be seriously considered as a sufficient evidence. We also mention that the whole line
of previous papers connected to ours (Agarwal et al. [1], Bach & Perchet [3], Duchi et al [13], Flaxman et al. [16]
Jamieson et al [20], Shamir [32]) presents only theoretical results.

**R2, R3:** *Give guarantees in the case when the function parameters are unknown.* This is an interesting question related
to adaptive techniques. To the best of our knowledge, it was not answered in the literature even for simpler settings
where $\beta = 1$ or $\beta = 2$ and one only needs to adapt to the Lipschitz constant $L$ and strong convexity parameter $\alpha$. When
the question is to find algorithms optimizing the bounds, all the previous related work assumes explicitly known $L$
(known $\alpha$ if the functions are strongly convex), cf. references above among others.

**R2, R4:** *Use of the kernel being redundant in the case $\beta = 2$.* Inspection of the proof of Lemma 2.3 shows that for
$\beta = 2$ the bias is of the same order even if $K = 1$ and the randomization in $r$ is suppressed (this is not true for bigger
$\beta$). For the variance term (Lemma 2.4), the dependency on $K$ does not influence the rate for all $\beta > 1$.

**R2:** *Strong convexity assumption.* This assumption is quite common in the bandit literature, cf. for example, Agarwal et
al. [1], Bach & Perchet [3], Hu et al. [17], Shamir [32]. Discussion of motivation for strong convexity assumption can
be found there. We did not reproduce this discussion due to space limitations.

**R2.** *Noise assumption in Theorem 6.1 (lower bound):* Using the second order expansion of the logarithm w.r.t. $v$, it is
not hard to check that this assumption is satisfied when $F$ has a smooth enough density with finite Fisher information.

**R2:** *Choice of the learning rate for the unconstrained case (Eq. (6)).* Having two regimes for tuning parameters is a
technical point in order to cover small enough strong convexity parameters $\alpha$. If we drop the first regime, i.e. set $T_0 = 0$
(start from $t = 1$) then we cannot guarantee that in the display below Eq. (34) the term with $\bar{L}^2/\alpha t$ on the r.h.s. is small.

**R2:** *Choice of strong convexity parameter in Theorem 3.2.* It is a natural assumption since the case of very small strong
convexity parameter boils down to the setting without strong convexity, where higher order smoothness does not help
and the optimal rate is different. This is discussed in Section 7.

**R2:** *Additional references on bandit literature:* Thanks for pointing out the papers on contextual bandit, which we
will cite in the revision. Dealing with the same Holder assumption on the arm' reward functions [Hu et al. 2019] their
setting is significantly different from ours as at each step the learner has to choose between finite number of arms, while
in our case it is a continuum. In the optimal rate for contextual bandits (see [Hu et al. 2019]) the dimension appears in
the exponent of $T$, whereas in our setting the dimension is only a factor.

**R2:** *line 95.* One verifies that for this choice of $K$, it holds that $\kappa_\beta \leq 3\beta^3$. We'll include this in the revision.

**R2:** *Line 430.* Indeed, in this argument the simplification comes by roughly dropping the quantities bounded by 1.

**R2:** *Summarized some of results in [3] in Appendix Section C.* In Appendix C we have summarized some results of [3]
by indicating the suboptimal rates that they obtain. Concerning the incorrect results, we have indicated where there is a
problem in the argument.

**R2:** *More discussion.* We will add the definition of cumulative regret in the main paper and give more intuition behind
the proofs whenever possible in the paper or in the appendix. Concerning the broader impact, as our work is theoretical,
we do not expect a broader impact, nor possibly negative effects on society. If the reviewers detected any relevant aspect
we missed, we are happy to fill this section.

**R3:** *I am unsure of the significance of this contribution.* We show what is the optimal choice of parameters for
the algorithm by Bach & Perchet and, notably derive near matching lower bounds. Moreover, we provide the first
polynomial time method of estimation of the minimum value $\min_x f(x)$ with near oracle behavior.

**R3:** *Related work section at the beginning of the paper.* We prefer to keep it at the end of the paper since it is easier to
compare to the previous literature once the results have been presented. However notice that in the introduction we
briefly comment on key related papers and improvements developed by our work.

**R4:** *Adding a table.* We have also thought about this option but did not choose it since there are not too many cases to
compare with.

**R4:** *Why generalization to "adversarial noise" is plausible?* The reason is that multiplication of $y_t$, $y_t'$ by $\zeta_t$, which is
zero-mean, makes the stochastic error zero-mean independently of the nature of the noise.

**R4:** *Lower bound - what is new?* The bound extends over the initial proof technique of Polyak and Tsybakov [28] by
accumulating multiple (rather than two) probe functions to account for the dependency on the dimension $d$ and $\alpha$ and
applying Assouad's Lemma to obtain the final result.

**R4:** *Why does def 2.1 need to constrain $\zeta$ and $r$?* There is no def 2.1, probably assumption 2.1? Indeed, informally it is
clear that $\zeta$ and $r$ should be chosen independently. We explicitly state this requirement in the assumption just to make
the theorems formally correct.

**R5:** *line 180 and 181:* by the fact that it cannot be improved we mean that even if $x^*$ is known and we are sampling $T$
times $f(x^*) +$ "noise", all at the same point $x^*$, the error is of the order $1/\sqrt{T}$. We will modify the discussion in the
revision, also referring to Thm. 4.1.

[Meta-Review · NeurIPS 2020]

Thank you for the rebuttal that satisfied all the reviewers in most of their questions, and there was an unanimous agreement that this is a high quality submission. The only remark (and reason why the scores are not higher) is that the authors should discuss more the significance of their results. I ask the authors to incorporate this in their revised version and I suggest to accept the submission.